# MultiTune: Phase-Aware Multi-Objective Tuning for Diffusion Models

## Abstract

Diffusion models excel at basic text-to-image but struggle to align with specific objectives. While reinforcement learning offers a promising solution, single-reward setups often lead to overfitting. To this end, multi-objective optimization methods are proposed. However, such methods face challenges of goal conflicts, inflexible reward fusion, and low efficiency, hindering overall performance across diverse criteria. To address these challenges, we propose MultiTune, a lightweight multi-objective framework tailored to the diffusion process. We decompose the optimization targets into Phase and Main objectives, where the former involves multiple phases of stepwise guidance and the latter ensures overall convergence. We first introduce a phase-aware switching strategy that aligns with the structural-to-textural evolution in diffusion, enabling dynamic and decoupled scheduling of Phase Objectives. Then, we adaptively balance the Phase and Main Objectives based on variations in image quality for on-demand collaboration. Finally, we cut costs via sample pruning and early stopping. Experiments demonstrate that MultiTune outperforms SOTA methods in aesthetics, semantics, details, and style, achieving leading performance across five quantitative metrics.

## 1 Introduction

Diffusion models (Rombach et al., 2022) have demonstrated impressive performance on the Text to Image (T2I) tasks (Ramesh et al., 2022; Betker et al., 2023; Clark et al., 2023; Lin et al., 2024; Esser et al., 2024; Lee et al., 2024a; Li et al., 2024; Yuan et al., 2024; Ding et al., 2025), but still face performance bottlenecks when optimizing specific criteria such as aesthetics and text-image alignment. Reinforcement learning (RL) has thus been introduced to directly optimize these downstream objectives (Li et al., 2022; Wu et al., 2023b;a; Zhang et al., 2024a; Deng et al., 2024; Karthik et al., 2024). However, most RL methods typically rely on sparse and single rewards, which are only available after the denoising process has been completed. This setup easily leads to reward hacking (Ibarz et al., 2018; Skalse et al., 2022), causes over-optimization of a single objective, and impairs the model's generalizability (Gao et al., 2023; Rafailov et al., 2024).

To alleviate the limitations associated with optimizing a single objective, recent efforts focus on incorporating multi-objective paradigms into diffusion models. Existing methods mainly include reward weighting (Gu et al., 2024; Zheng & Wang, 2023; Van Moffaert et al., 2013) and model soup (Yang et al., 2024b; Lee et al., 2025).

Reward weighting methods (Agarwal et al., 2022) optimize a unified objective by linearly combining different reward functions into a single scalar signal. These methods are cost-efficient and allow for flexibly prioritizing the objectives. However, fixed weights may fail to adapt to changing objective importance, causing overfitting specific objectives while neglecting others (Lee et al., 2025). The weighting coefficients are often determined based on manual experience or expensive hyperparameter tuning, which constrains the scalability and generalizability of these methods in complex tasks with multiple objectives (Xia et al., 2021). More importantly, these methods are agnostic to the dynamics of diffusion models, and the compounded rewards provide sparse and weak supervision for the optimization process. The potential inner conflicts make the optimization ineffective and unstable.

Another approach, model soup, trains sub-models on each objective separately and fuses their parameters at inference time (Rame et al., 2023; Wortsman et al., 2022). Nevertheless, the parameter fusion is still based on static weights, making it difficult to respond to the evolving nature of the

generation process dynamically. This is particularly problematic in diffusion models, where generation progresses through distinct stages and requires flexible adjustment of optimization strengths. Moreover, multiple models must be trained, incurring high computational cost (Yang et al., 2024b), and parameter fusion may lead to semantic drift or performance drops, affecting output consistency and controllability (Gao et al., 2023; Rafailov et al., 2024).

Despite early gains, multi-objective methods still face three core challenges in T2I. First, it is difficult to balance diverse objectives without dynamic adaptation. Second, without aligning with the progressive nature of diffusion, existing methods fail to control objectives across stages robustly (Choi et al., 2022; Li et al., 2023; Yi et al., 2024; Xie & Gong, 2025). Third, they suffer from sparse rewards during denoising, hindering effective and stable optimization. *A detailed review of **Related Works** is provided in Appendix A.*

To overcome the above limitations, we propose MultiTune, an adaptive multi-objective framework aligned with the inherent generation progress (see Fig. 1) of diffusion models—from coarse structure to fine texture. Specifically, we first introduce a phase-aware switching strategy that selectively activates task-relevant objectives at each generation phase. To address sparse rewards, MultiTune introduces intrinsic rewards from consecutive image changes, providing stable feedback for exploration. While intrinsic signals encourage exploration, Phase Objectives guide it, together improving both learning efficiency and exploration precision. Secondly, we design a trend-based adaptive mechanism that adjusts the contribution between Phase and Main Objectives by tracking quality changes across

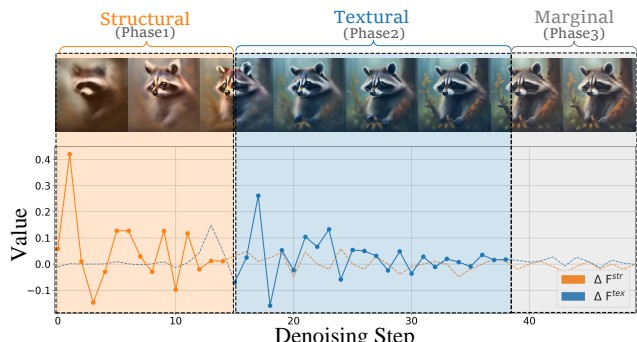

Figure 1: Illustration of different phases of the denoising process. 1) Early stage focuses on forming the image's overall structure, leading to drastic changes of the structural score ($\Delta F^{\mathrm{str}}$). 2) Textural enhancement becomes dominant after structure formation, reflected by the sudden increase in textural score changes ($\Delta F^{\mathrm{tex}}$). 3) The optimization of both structure and texture becomes marginal in the late stage.

timesteps. In addition, we propose an efficiency-aware module that reduces computation by discarding unhelpful samples and truncating redundant generation steps upon convergence. *Finally, we provide a detailed discussion of the significant differences between MultiTuneand existing methods in Appendix. A.4.*

We evaluate MultiTune via extensive experiments covering multi-objective evaluation, mechanism ablation, and transfer generalization. Across diverse T2I tasks, MultiTune outperforms all baselines, improving generation quality and significantly reducing computational cost, confirming its effectiveness and adaptability. Experiments demonstrate our leading performance across five quantitative metrics. Our contributions are:

- We propose a multi-objective switching strategy that aligns with the diffusion process from structure to detail, dynamically focusing on phase-specific targets for decoupled and ordered optimization.

- We propose an adaptive mechanism based on image quality changes, which dynamically balances between Phase Objectives and Main Objective during training, enabling real-time shift of focus.

- We achieve lightweight multi-objective training by discarding ineffective samples and early stopping redundant denoising steps.

## 2 METHOD

To achieve multi-objective adaptive coordination and efficient fine-tuning for diffusion models, we propose MultiTune, which establishes a hierarchical system guided by Phase Objectives for local optimization and the Main Objective for global convergence. In Sec. 2.2, we design a phase-aware

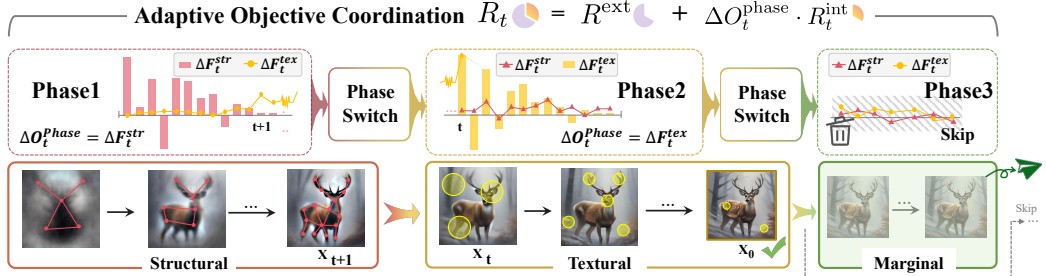

Figure 2: Overview of MultiTune. MultiTune involves a phase-aware intrinsic rewards that aligns with the progressive nature of the denoising process, which guide training from structural construction to detail refinement. To balance the importance of Phase and Main objectives, an adaptive coordination mechanism is employed by tracking changes in image quality. To reduce unnecessary computation, an early stop mechanism is applied according to both the sample and convergence levels.

switching mechanism that leverages the progressive nature of diffusion models to guide training from structural construction to detail refinement. In Sec. 2.3, we introduce an adaptive coordination mechanism that dynamically balances the importance of different objectives by tracking image quality changes. In Sec. 2.4, we propose a screening mechanism from both the sample and convergence levels to reduce computational overhead. The framework of MultiTune is illustrated in Fig. 2.

## 2.1 RL FOR DIFFUSION MODEL

**MDP for Diffusion Models:** We formulate the denoising process of diffusion models (Ho et al., 2020; Song et al., 2020) as a Markov Decision Process (MDP), represented as a tuple: $\langle \mathcal{S}, \mathcal{A}, P, \rho_0, R, T \rangle$, where $\mathcal{S}$ is the state space containing all intermediate images and $\mathcal{A}$ denotes the action space, where each action corresponds to the predicted noise at a given step. The optimization goal is to maximize the expected cumulative reward based on generation quality via RL. *MDP terms and RL formalism are in the Appendix. B.*

**Multi-objective:** Our objectives include Phase Objectives ($O_t^{\text{phase}}$) and Main Objective ($O_t^{\text{main}}$). $O_t^{\text{phase}}$ shifts from structure to detail during denoising. $O_t^{\text{main}}$ targets specific task metrics like aesthetics, and will be optimized throughout the image generation process. *More details in Appendix C.* *It is important to note that the Phase Objectives and the Main Objective proposed in this paper do not rely on any specific reward model; they can flexibly incorporate or combine different reward formulations according to task requirements.*

**Intrinsic Motivation:** With intrinsic rewards (Burda et al., 2018; Badia et al., 2020; Ladosz et al., 2022; Chen et al., 2025), the MDP extends to $\langle \mathcal{S}, \mathcal{A}, P, \rho_0, R^{\text{ext}}, R^{\text{int}}, \lambda, T \rangle$, where $R^{\text{ext}}$ is the extrinsic reward, and $R^{\text{int}}$ provides stepwise feedback to boost exploration. When $R^{\text{ext}}$ is sparse, $R^{\text{int}}$ fills the feedback gap. The total reward is:

$$R = \lambda \cdot R^{\text{int}} + R^{\text{ext}}, \quad (1)$$

where $\lambda \in [0, 1]$ controls intrinsic reward strength. In RL, it is often manually fixed, limiting adaptive shift between Phase (intrinsic) and Main (extrinsic) Objective. Sec 2.3 introduces an adaptive factor to replace the static $\lambda$.

## 2.2 PHASE-AWARE OBJECTIVE SCHEDULING

Recent studies have shown that incorporating multiple objectives in fine-tuning helps improve T2I quality (Lee et al., 2024b; Hao et al., 2023). However, mixed optimization often causes conflicts between objectives, hindering their individual effectiveness. Coordinated optimization requires carefully scheduling their timing and contributions. Fig. 1 illustrates the phase nature of generation: in Phase 1, significant fluctuations in structural metrics indicate that early denoising focuses on global skeleton construction; in Phase 2, structural metrics stabilize while textural metrics become active, reflecting ongoing refinement of detail and style; in Phase 3, both metrics converge, and further

generation yields limited gains. These metric dynamics reveal phase-specific sensitivity to objectives, naturally leading to temporal decoupling. Based on this observation, we propose a phase-aware scheduling mechanism that aligns with the structural-to-textural transition in generation, selectively activating the most relevant objective at each phase to achieve temporally decoupled and orderly multi-objective switching.

To implement this strategy, we split the denoising trajectory $\{\mathbf{x}_t\}_{t=0}^T$ into three phases: structural construction $\mathcal{P}_{\text{Structural}}$, textural enhancement $\mathcal{P}_{\text{Textural}}$, and marginal optimization $\mathcal{P}_{\text{Marginal}}$. Each phase targets a distinct aspect of generation—global skeleton, fine-grained detail, and diminishing returns, respectively. Based on the above division, we can construct a new multi-objective learning paradigm, which optimizes the corresponding $O_t^{\text{phase}}$ at different phases by the inherent denoising nature of diffusion models.

Specifically, in the structural construction phase, we set the objective $\Delta F_t^{\text{str}}$ guided by the discrimination of the text-image alignment function, aiming to ensure semantic consistency and overall structural skeleton, which is defined as:

$$\Delta F_t^{\text{str}} = F^{\text{str}}(\hat{\mathbf{x}}_0(\mathbf{x}_t)) - F^{\text{str}}(\hat{\mathbf{x}}_0(\mathbf{x}_{t+1})), \tag{2}$$

where, $F^{\text{str}}(\cdot)$ is the reward function that measures structural alignment consistency, and $\hat{x}_0(x_t)$ is the ground-truth sample inferred directly from the denoising result at step $t$.

In the detail enhancement phase, the objective is switched to the detail- and style-oriented target $\Delta F_t^{\text{tex}}$, which emphasizes that the image textural matches local expressiveness, defined as:

$$\Delta F_t^{\text{tex}} = F^{\text{tex}}(\hat{\mathbf{x}}_0(\mathbf{x}_t)) - F^{\text{tex}}(\hat{\mathbf{x}}_0(\mathbf{x}_{t+1})), \tag{3}$$

where $F^{\text{tex}}(\cdot)$ denotes the reward function that discriminates texture quality preferences.

Therefore, the variation of the Phase Objectives $O_t^{phase}$ can be defined as:

$$\Delta O_t^{\text{phase}} = \begin{cases} \Delta F_t^{\text{str}} & \text{if } t \in \mathcal{P}_{\text{Structural}} \\ \Delta F_t^{\text{tex}} & \text{if } t \in \mathcal{P}_{\text{Textural}} \\ \text{skip} & \text{if } t \in \mathcal{P}_{\text{Marginal}} \end{cases}, \tag{4}$$

where skip denotes terminating the current sample optimization by skipping the remaining steps after entering $\mathcal{P}_{\text{Marginal}}$ (see Sec. 2.4 for details).

In T2I tasks, due to the heterogeneity of diffusion processes guided by different prompts, the phase length during generation also varies accordingly. To avoid the rigidity of phase partitioning based on human heuristics, we introduce a dynamic phase switching mechanism. Specifically, when the Phase Objectives scores converge with negligible and stable changes across timesteps, the condition for phase switching is met. Formally, we first define the variance of Phase Objectives $\Delta^2 O_t^{\text{phase}}$ as follows.

$$\Delta^2 O_t^{\text{phase}} = \Delta O_t^{\text{phase}} - \Delta O_{t+1}^{\text{phase}}. \tag{5}$$

Then, the indicator for phase transition $\mathbb{I}$ can be defined as:

$$\mathbb{I}^{\text{tran}} = (\Delta O_t^{\text{phase}} \to 0) \& (\Delta^2 O_t^{\text{phase}} \to 0). \tag{6}$$

When $\mathbb{I}^{\text{tran}} = 1$, $\Delta O_t^{\text{phase}}$ is switched to the next phase according to Eq. 4. We illustrate the implementation of $\to 0$ in Appendix D.1.

## 2.3 OBJECTIVES TUNE AND ADAPTIVE COORDINATION

Current multi-objective methods often use static weighting, which lacks adaptability and transferability. This commonly causes objective bias by over-optimizing high-weighted objectives while neglecting others (Lee et al., 2024b). To meet the urgent need for adaptive coordination, we propose a quality-driven mechanism that balances Phase and Main Objectives. This design adjusts the weighting of intrinsic and extrinsic rewards along the diffusion timeline: in early stages, intrinsic rewards dominate to enhance intermediate feedback and focus on Phase Objectives optimization; as denoising progresses and the image forms, the weight of intrinsic rewards is reduced while extrinsic rewards are emphasized, smoothly shifting the optimization focus toward the Main Objective and ensuring stable policy convergence.

**Intrinsic Reward:**   We adopt intrinsic rewards to alleviate sparse rewards in existing approaches. Intrinsic rewards in MultiTune serve two purposes: first, to encourage exploration, enhance the diversity of the generated distribution, and alleviate reward hijacking; second, to enable the optimization of Phase Objectives. Specifically, we define the intrinsic reward as:

$$R_t^{\text{int}} = \|\hat{\mathbf{x}}_0(\mathbf{x}_t) - \hat{\mathbf{x}}_0(\mathbf{x}_{t+1})\|_2^2. \tag{7}$$

By introducing $R_t^{int}$, the model is encouraged to explore diverse denoising trajectories, thereby preventing premature convergence to a single synthesis mode and enhancing the generation diversity.

**Mutual Boost:**   After constructing intrinsic rewards, we use the change in Phase Objectives $\Delta O_t^{\text{phase}}$ to guide the exploration behavior by evaluating whether the current exploration leads to improvements in the Phase Objectives, thereby ensuring the rationality of the exploration direction. Note that the value of $\Delta O_t^{\text{phase}}$ can be both positive and negative, with positive values indicating the promotion of intrinsic reward exploration, and negative values representing penalization. As such, $\Delta O_t^{\text{phase}}$ and $R_t^{\text{int}}$ form a synergistic feedback loop: $R_t^{\text{int}}$ drives generation diversity while promoting the optimization of Phase Objectives; meanwhile, $\Delta O_t^{\text{phase}}$ serves as value feedback to guide exploration towards directions beneficial for the improvement of Phase Objectives. This mechanism maintains dynamic consistency between exploration behavior and objective improvement by coupling exploration potential with Phase Objectives gains.

**Adaptive Synergy:**   Specifically, we use $\Delta O_t^{phase}$ as a balancing coefficient to adjust the weighting dynamically. This mechanism encourages diverse exploration and strengthens Phase Objectives during early steps of denoising, while gradually shifting the optimization focus towards the predefined downstream tasks in later steps of denoising, guiding the generated results to converge to a distribution aligned with the Main Objective stably. Finally, the overall reward function is defined as:

$$R_t = \underbrace{\Delta O_t^{\text{phase}} \cdot R_t^{\text{int}}}_{\text{Phase Reward}} + \underbrace{R^{\text{ext}}}_{\text{Main Reward}}. \tag{8}$$

We eliminate the manually set fixed parameter $\lambda$ in Eq. 1 and instead introduce the adaptively varying $\Delta O_t^{\text{phase}}$ to achieve dynamic weight adjustment and efficient coordination between Phase Objectives and cooperative objectives.

$$R^{\text{Phase}} = \Delta O_t^{\text{phase}} \cdot R_t^{\text{int}} \tag{9}$$

## 2.4 Efficiency-Aware Training Optimization

Previous studies show that RL fine-tuning T2I models offers strong optimization ability but incurs high computational cost (Black et al., 2023; Fan et al., 2023; Yang et al., 2024a). To address this, we introduce efficiency-oriented mechanisms that reduce redundant computation at both the sample and step levels, achieving high generation quality with lower resource usage.

**Sample Level:**   First, despite both positive and negative $\Delta O_t^{\text{phase}}$ contributing to the regularization of intrinsic reward's exploration, negative rewards may still exert excessive penalization on the policy. Therefore, we apply a dynamic filtering strategy: If $\Delta O_t^{\text{phase}} \leq 0$, this status may be partially discarded to reduce gradient noise and improve training efficiency. Formally, it could be written as:

$$\Delta O_t^{\text{phase}} = \mathbb{I}_t^{\text{disc}} \Delta O_t^{\text{phase}} \quad \text{if} \quad \Delta O_t^{\text{phase}} \leq 0, \quad \mathbb{I}_t^{\text{disc}} \sim \text{B}(p), \tag{10}$$

where $\mathbb{I}_t^{\text{disc}}$ is the indicator for filtering and $\text{B}(\cdot)$ denotes the Bernoulli distribution.

**Step Level:**   To avoid over-optimizing stabilized regions in later denoising, we introduce a dynamic skipping mechanism based on both the change rate and trend of the Textural reward. Unlike static thresholds, it adaptively decides early stopping when the reward shows convergence. Since the Textural reward already integrates key factors like structural fidelity and aesthetics, its stabilization implies diminishing returns or potential artifacts(See the gray region in Fig. 1). When $t \in \mathcal{T}_{\text{Marginal}}$, denoising and reward computation are terminated, and the final trajectory becomes $\{\mathbf{x}_t\}_{t=T}^0$. Please refer to the Alg. 1 in the *Appendix* for a clear understanding of the proposed skipping strategy.

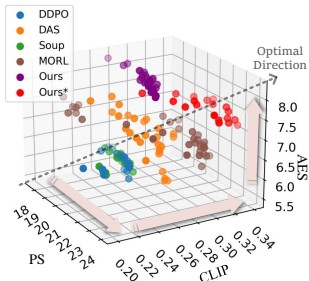

Figure 3: Multi-objective Pareto optimal front (3-D) on SDv21: our method leads in all objectives.

Table 1: Comparisons of multi-objective optimization using averaged metrics on two popular backbones. Best is highlighted in bold.

| Method | SDv15 | | | | | SDv21 | | | | |
|---|---|---|---|---|---|---|---|---|---|---|
| | AES | CLIP | PS | TCE | LPIPS | AES | CLIP | PS | TCE | LPIPS |
| DDPO | 6.897 | 0.301 | 21.60 | 37.32 | 0.616 | 6.178 | 0.259 | 18.51 | 38.63 | 0.649 |
| Soup | 6.754 | 0.303 | 21.69 | 39.27 | 0.634 | 6.173 | 0.262 | 18.77 | 38.98 | 0.657 |
| MORL | 6.875 | 0.310 | 21.28 | 37.92 | 0.651 | 6.912 | 0.244 | 21.32 | 34.71 | 0.589 |
| DAS | 6.579 | 0.293 | 20.95 | 39.21 | 0.656 | 6.577 | 0.242 | 20.90 | 39.93 | 0.657 |
| Ours | **7.419** | 0.313 | 21.89 | 37.95 | 0.612 | **7.926** | 0.275 | 20.02 | **40.10** | **0.690** |
| Ours* | 7.136 | **0.318** | **22.01** | **41.05** | **0.667** | 7.529 | **0.281** | **21.90** | 39.94 | 0.681 |

By jointly optimizing sample selection and training steps, MultiTune achieves superior multi-objective performance with significantly lower computational cost.

# 3 EXPERIMENTS

## 3.1 SETUP

**Datasets.** To compare with previous studies, we use the same prompt set consisting of 45 different animal classes for training (Simple-animals), which has been widely adopted for reward fine-tuning. For testing, we generate 8 different noises for each animal class, totaling 360 test samples. For generalization evaluation, larger-scale prompt sets are further considered for testing, including 398 animal classes from ImageNet (ImageNet-A) and a subset of the HPSv2 dataset with 500 complex prompts (HPSv2-S). *In addition, to verify that our method remains effective on larger and more complex datasets, we further report experimental results on datasets such as Pick-a-Pic and GenEval (see Fig 14).*

**Rewards and Metrics.** In this paper, we mainly take Aesthetic Score (AES) (Schuhmann et al., 2022a) (Main Objective), Clip Score (Structural Objective, CLIP) (Radford et al.), and PickScore (Textural Objective, PS) (Kirstain et al., 2023) as the multi-objectives. We also introduce ImageReward (IR) (Xu et al., 2023) as the supplementary target to further analyze the effectiveness of

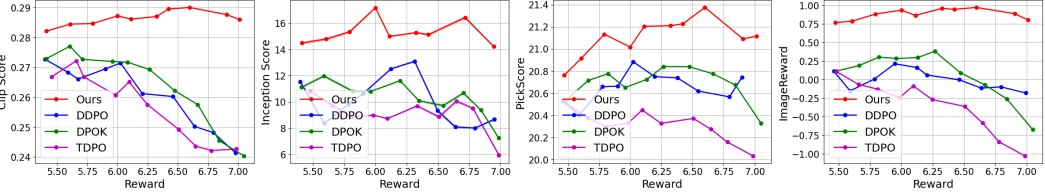

Figure 4: Cross-metric performance compared to methods with fixed Main Objective.

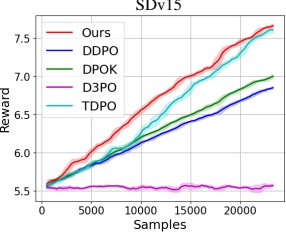
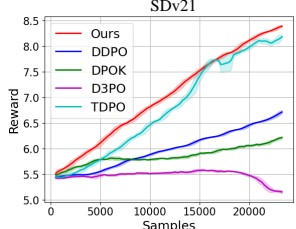
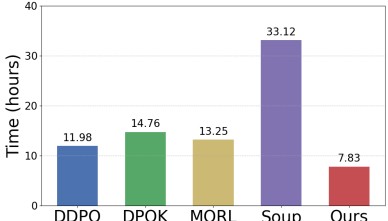

Figure 5: Learning performance of enhancing the Main objective compared with single-objective methods.

Figure 6: Computational cost to reach Aesthetic Score=7.

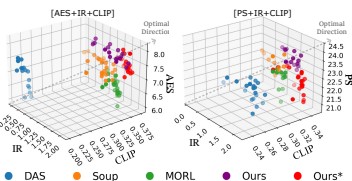

Figure 7: Alternative reward functions as objectives.

Table 2: Overall ablation study. Time refers to the computational cost for single epoch (minutes). Mem refers to peak GPU memory consumption (GB).

| Ablation Variant | AES | CLIP | PS | Time | *Mem* | Top2 |
|---|---|---|---|---|---|---|
| $[R_{int}]$ | 7.271 | 0.305 | 21.09 | **6.493** | 52.06 | 2 |
| $[R_{int}(\Delta F^{\text{str}} + \Delta F^{\text{tex}})]$ | 7.109 | 0.307 | 21.64 | 9.930 | 60.40 | 0 |
| $[R_{int}\Delta F^{\text{str}}, R_{int}\Delta F^{\text{tex}}]$ | 7.226 | 0.310 | **22.42** | 8.799 | 62.21 | 2 |
| $[R_{int}\Delta F^{\text{str}}, R_{int}\Delta F^{\text{tex}}, \text{skip}]$ | **7.419** | **0.313** | 21.89 | 6.769 | 56.18 | 4 |

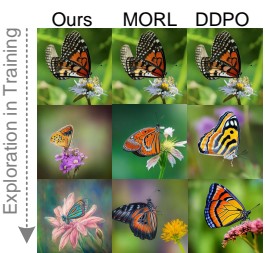 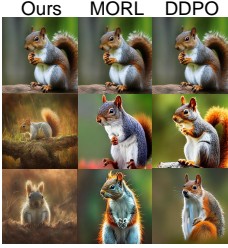 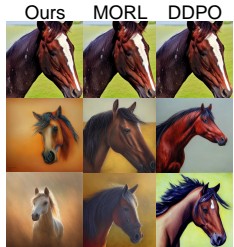 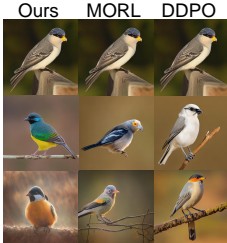

Figure 8: Exploration capability over training. With intrinsic rewards, MultiTune continues to explore by varying structure, color, and background. In contrast, baseline methods show limited exploration.

MultiTune (see Sec. 3.4.2). The aforementioned rewards are also treated as the evaluation metrics, and we further deploy Inception Score (IS) (Salimans et al., 2016), TCE (Ibarrola & Grace, 2024), and LPIPS (Zhang et al., 2018) to assess image diversity.

**Baselines.** For multi-objective tuning, we introduce three advanced baselines: MORL (Yang et al., 2019), Soup (Rame et al., 2023), and DAS (Kim et al., 2025) with MORL. Then, we compared with the popular single-objective methods targeting aesthetic score to discuss the improvement of the Main Objective, including DDPO (Black et al., 2023), DPOK (Fan et al., 2024), D3PO (Yang et al., 2024a), and TDPO (Zhang et al., 2024b). We deploy Stable Diffusion v1.4 (SDv14), Stable Diffusion v1.5 (SDv15), v2.1-turbo (SDv21), and XL1.0 (XL) as the backbones in experiments. *Furthermore, to demonstrate that our method remains effective for larger and more advanced diffusion models, we additionally conduct both qualitative and quantitative experiments on the SD3 model (see Fig. 15 and Tab. 7). To this end, we verify the applicability of MultiTuneon diffusion models based on both U-Net and DiT architectures, demonstrating its broad effectiveness across different diffusion model paradigms.*

*Reward Model Combination. To verify that the proposed Phase Objectives and Main Objective do not rely on any specific reward model and can flexibly incorporate different reward formulations based on task requirements, we further conduct experiments that replace the reward models. These experiments evaluate the applicability and robustness of MultiTuneunder various reward configurations.*

*Experimental List: In summary, to fully validate the effectiveness of MultiTune, this paper provides as many as 21 experiments of different types. The detailed list of experiments can be found in Appendix. G.*

## 3.2 MULTI-OBJECTIVE OPTIMIZATION

In Fig. 3, we present the three-dimensional Pareto optimal fronts of different methods. See its 2-D projections in the *Appendix*. To provide a quantitatively intuitive demonstration, we also report the average evaluation results of these methods across two backbones in Tab. 1. Since MultiTune can also incorporate combined objectives as the Main Objective, we design **Ours\*** with multi-objective as $O^{main}$, following the protocol used in MORL. It can be observed that employing MultiTune alone already surpasses existing approaches, and the performance is further enhanced by adopting Ours\*.

## 3.3 COMPARING WITH SINGLE-OBJECTIVE METHOD

Here, we compare our method with single-objective approaches in terms of optimization effectiveness on the Main Objective (*i.e.*, aesthetic score). Firstly, as shown in Fig. 5, the aesthetic scores of our

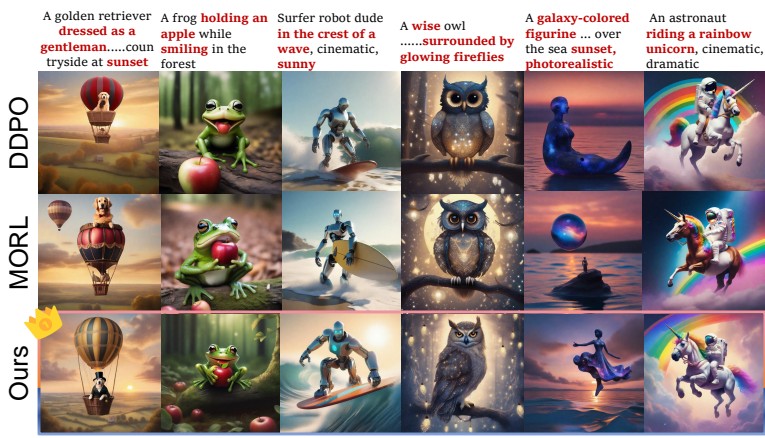 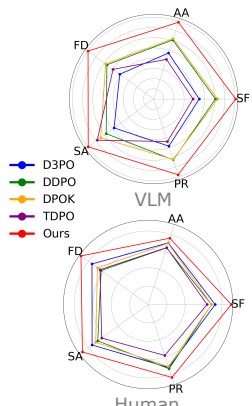

Figure 9: Overall visual quality of generated images. Our results achieve superior aesthetic impressions with finer texture and more precise alignment to the semantic elements.

Figure 10: Subjective Evaluation by both VLM and Human.

method are learned more effectively on both SDv15 and SDv21, indicating that MultiTune also fosters improved Main objective learning efficiency by leveraging multi-objectives in a decoupled manner. Subsequently, Fig. 4 illustrates the performance of different methods across multiple evaluation metrics under different levels of the same Main Objective scores. Our method significantly surpasses existing single-objective methods across all four metrics.

## 3.4 ANALYSIS OF MULTITUNE

**Overall Ablation.** MultiTune is a three-phase framework comprising structural optimization, textural optimization, and a marginal phase skipping. By taking our method as three phases, we design three ablation variants, including: 1) excluding the marginal phase (two-phase), 2) concurrently optimizing structure and texture without phase-wise decomposition (single-phase), and 3) introducing only intrinsic rewards *without* structural and textural considerations (zero-phase). As shown in Tab. 2, we fine-tune on SDv15 to compare the multi-objective performance and time cost across these variants. It can be observed that our proposed method achieves optimal multi-objective performance within an acceptable computational cost.

**Analysis of using alternative functions as objectives.** To validate the generalizability of our approach concerning objective functions, we investigate substituting the current optimization objectives with alternative reward functions possessing similar characteristics. Specifically, we replace the Phase Objectives (originally PS in Phase 2) with IR to analyze the impact of substituting the Phase Objectives, denoted by [AES+IR+CLIP] Subsequently, we further substitute the Main Objective (originally AES) with PS, denoted by [PS+IR+CLIP]. Correspondingly, we perform the same replacements of objective functions for comparative multi-objective methods as baselines. As shown in Fig. 7, we compare the resulting 3-D Pareto optimal fronts across these modified objectives on SDv15. It can be observed that our method consistently achieves optimal performance across multiple dimensions for these objectives, thus substantiating its universal efficacy across various objective functions. In Appendix, we also replace current CLIP with other alignment reward functions (LAION-C and ALIGN), and the results demonstrate our generalizability.

**Computational Cost.** Under identical device settings and configurations, we compared the time consumption of different methods when the main aesthetic objective reaches 7. As shown in Fig. 6, both single-objective and multi-objective approaches consume considerably more time compared to our method. *Moreover, Tab. 2 presents the computational cost and memory cost of different ablation variants, which reveals the detailed mechanism behind the superior efficiency of MultiTune. The results show that introducing $\Delta F$ increase about 10GB cost, while skipping the Phase3 can reduce the gradient accumulation step per sample, thus saving memory.*

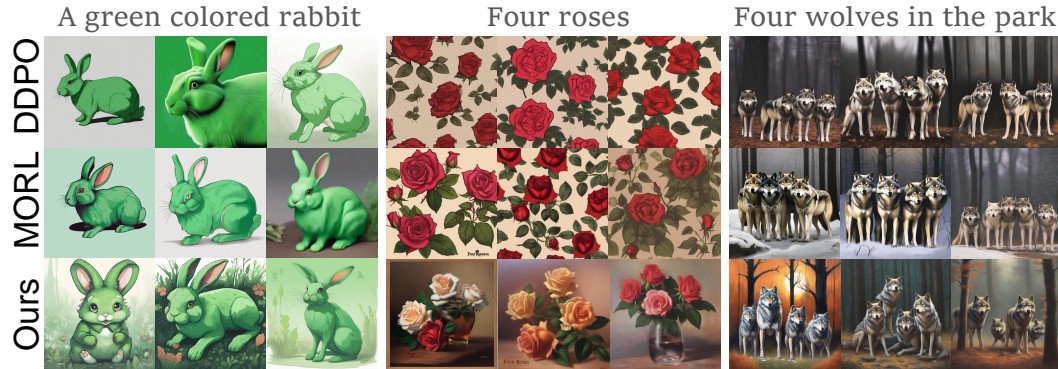

Figure 11: Diversity of the generated images. With high prompt-image consistency, our final generated images exhibit improved diversity in structure, color, and background.

Table 3: *Testing* on complex prompt datasets.

| Method | HPSv2-S | | | ImageNet-A | | |
|---|---|---|---|---|---|---|
| | AES | CLIP | PS | AES | CLIP | PS |
| DDPO | 5.69 | 0.274 | 18.51 | 6.66 | 0.271 | 20.75 |
| MORL | 5.68 | **0.283** | 19.69 | 6.51 | 0.285 | 20.63 |
| Ours | **5.97** | 0.281 | **20.19** | **7.13** | **0.287** | **21.89** |

Table 4: *Training* on complex prompt datasets.

| Method | HPSv2-S | | | ImageNet-A | | |
|---|---|---|---|---|---|---|
| | AES | CLIP | PS | AES | CLIP | PS |
| DDPO | 6.00 | 0.281 | 20.14 | 6.99 | 0.302 | 22.15 |
| MORL | 5.89 | **0.290** | 20.77 | 6.72 | 0.303 | 22.08 |
| Ours | **6.45** | 0.286 | **21.89** | **7.29** | **0.311** | **22.97** |

## 3.5 VISUAL IMPRESSIONS OF GENERATION QUALITY

**Overall Generative Quality.** In Fig. 9, we present generated images with complex prompts to discuss the overall generative quality. It can be observed that our generated results achieve superior alignment with the critical semantic elements in the prompts, while also exhibiting enhanced aesthetic style and finer image textures.

**Diversity and Alignment.** To investigate visual diversity and alignment, we also provide images with the same prompt, checkpoint, and different seeds to show generative diversity. In Fig. 11, the results demonstrate the superior alignment, diversity, and quality of our generated images.

**Exploration.** To simultaneously demonstrate the diversity, visual quality, and multi-objective exploration capability of the proposed method, we design the experiment presented in Fig. 8. Specifically, we generate images using identical prompts and fixed random seeds at different training epochs. The existing methods always perform the same image content (*e.g.*, structure, color, and background) with limited aesthetic quality during training. In contrast, the continuous variation in image content illustrates both the diversity and the exploratory capacity of our method.

## 3.6 TRANSFER STUDY

To evaluate the transferability of MultiTune, we conducted experiments using a complex unseen prompt set and a broader range of backbone architectures. We employed both a classical single-objective method (DDPO) and a multi-objective method (MORL) as baselines for comparison.
**Complex Prompt Set.** Here, we conduct experiments on two more challenging prompt sets, namely ImageNet Animal (ImageNet-A) and a subset of HPSv2 (HPSv2-S). In Tab. 3, we first test simple-animal-trained models on these datasets to evaluate their effectiveness on unseen prompts. Although there is an inevitable performance drop on the unseen prompt set, our method still achieves the best results. In Tab. 4, we then take these prompts for training, and our method consistently exhibits improved performance.

**More SD Backbones.** Besides SDv1.5 and SDv2.1, we also conducted multi-objective experiments on both SDv1.4 and the more advanced SD-XL. As shown in Tab. 5, our superior results demonstrate that MultiTune can consistently enhance the fine-tuning performance of diverse diffusion backbones.

*Notably, the inferior performance of RL SDXl compared to SDv14 needs further discussion. This could be because SDXL has significantly more parameters and greater robustness than SD14, making it less sensitive to external signals, which leads to slower fitting of the reward function. In fact, we found that by increasing the batch size and extending the training time, SDXL can achieve higher reward scores. This further suggests that SDXL has a slower and more cautious learning efficiency, which is closely tied to its stability coming from the larger parameter size and the scale of its pretraining data.*

### 3.7 SUBJECTIVE EVALUATION

Table 5: Transferability to more backbones.

| Method | SDv1.4 | | | SD-XL | | |
|--------|--------|------|------|-------|------|------|
| | AES | CLIP | PS | AES | CLIP | PS |
| *Base* | 5.49 | 0.301 | 21.19 | 5.67 | 0.306 | 21.34 |
| DDPO | 6.67 | 0.298 | 21.36 | 5.74 | 0.286 | 20.35 |
| MORL | 6.52 | 0.310 | 21.70 | 5.95 | **0.307** | 21.28 |
| Ours | **7.25** | **0.314** | **22.26** | **6.45** | 0.305 | **22.16** |

To assess the perceptual quality of generated images, we conducted two complementary evaluations: Human Impression Assessment and Large Vision Model (LVM) Scoring. Using the fine-tuned SD-XL model, we generated images from HPSv2 prompts across several methods. Evaluation was based on five criteria—Structural Faithfulness (SF), Aesthetic Appeal (AA), Fine-grained Detail (FD), Semantic Alignment (SA), and Prompt Responsiveness (PR)—each independently rated by human participants and ChatGPT. Additional details of the subjective evaluation are provided in the *Appendix*. As shown in Fig. 16, our approach consistently outperforms others across both evaluation modalities and all metrics, which indicates the overall superior image quality of our method.

## 4 CONCLUSION

This paper presents MultiTune, a lightweight framework for multi-objective fine-tuning of diffusion models, combining phase-aware switching, adaptive weighting, and efficiency optimization. It effectively resolves objective conflicts, sparse rewards, and high costs, and achieves superior performance with strong generalization. In summary, MultiTune leverages adaptive strategies to align with the natural progression of diffusion models—from structural construction to detail refinement—achieving a unified framework for multi-objective alignment and efficient optimization.

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

# APPENDIX

# A ADDITIONAL RELATED WORKS

## A.1 FINE-TUNING DIFFUSION MODELS WITH RL:

Diffusion models have achieved remarkable progress in image generation tasks in recent years (Esser et al., 2024; Lee et al., 2024a; Li et al., 2024; Yuan et al., 2024; Ding et al., 2025). With their step-by-step denoising mechanism, they show strong performance across various evaluation metrics. However, their original training objectives mainly focus on reconstruction likelihood or data fitting, making it difficult to capture subjective preferences or high-level semantic features required by specific downstream tasks (Black et al., 2023). As a result, the research on improving task-specific generation quality is still inadequate. To bridge this gap, researchers have gradually introduced the reinforcement learning paradigm (Black et al., 2023; Fan et al., 2023; Yang et al., 2024a; Wallace et al., 2024) into the fine-tuning stage of diffusion models. The goal is to use external rewards or preference feedback to guide the model policy toward specific task objectives, thereby enabling explicit optimization for downstream performance.

Although early results show that RL can enhance target alignment and subjective quality, this direction still faces several challenges (Franceschelli & Musolesi, 2024). First, RL often relies on reward signals that are only available after the full image is generated, leading to sparse and delayed rewards. This can easily cause reward hacking or over-optimization. Second, RL fine-tuning typically introduces significant computational overhead, making it hard to balance performance gains with training efficiency. More importantly, the long-standing exploration-exploitation dilemma caused by RL is still unresolved. This trade-off becomes even more difficult in diffusion models, where generation is a long-sequence decision-making process, further intensifying the challenge of balancing exploration and stability (Franceschelli & Musolesi, 2024).

## A.2 INTRINSIC REWARD:

Intrinsic reward mechanisms (Burda et al., 2018; Badia et al., 2020; Ladosz et al., 2022; Chen et al., 2025) are commonly used in reinforcement learning to handle sparse feedback. They construct exploration signals based on state visitation counts or network prediction errors, providing optimization guidance for intermediate training steps. This helps improve the efficiency and diversity of policy learning. In diffusion models, different timesteps correspond to different semantic levels and have varying impacts on the final image. Under sparse reward conditions, it is difficult to evaluate the specific contribution of each denoising step. As a result, ineffective or even harmful strategies may be retained without timely correction, reducing optimization efficiency (Franceschelli & Musolesi, 2024; Hu et al., 2025). To address this, we introduce intrinsic rewards as optimization signals for intermediate denoising steps to encourage policy exploration and improve generation diversity.

## A.3 MULTI-OBJECTIVE OPTIMIZATION

Most existing RL fine-tuning methods rely on single-objective optimization, which often leads to overfitting to a specific reward and harms the model's generalization across other quality dimensions. To mitigate this issue, multi-objective optimization has gained increasing attention. Effectively balancing multiple image quality objectives has become a core challenge in building high-quality T2I generation systems. Prior works mainly explore two directions to address this challenge: reward-weighted MORL and model-based ensemble approaches. The first type, such as MORL (Agarwal et al., 2022; Gu et al., 2024; Zheng & Wang, 2023; Van Moffaert et al., 2013), combines different objectives via linear reward weighting to enable joint optimization. Although simple to implement, this strategy depends on heuristic or search-based weight selection. As the number of objectives grows, the cost increases sharply, and training becomes unstable due to gradient interference and objective conflicts. The second type, exemplified by Reward Soup (Yang et al., 2024b; Lee et al., 2025; Rame et al., 2023; Wortsman et al., 2022), trains separate sub-models aligned to each reward independently, and merges them through weight-based ensembling at inference. This approach avoids conflicts during joint training and improves stability and controllability. However, it requires training multiple models, resulting in high computational overhead. Inappropriate weight fusion may also introduce parameter inconsistency, semantic drift, or performance degradation.

### A.4 DIFFERENCES BETWEEN MULTITUNEAND EXISTING METHODS

#### A.4.1 DISTINCTIONS BETWEEN MULTITUNE'S MULTI-OBJECTIVE FRAMEWORK AND PRIOR MULTI-OBJECTIVE TECHNIQUES

*Notably, both categories of prior methods rely on static weighting for target integration, either at the reward level or the model level, and ignore the evolving nature of diffusion generation. Such static fusion fails to adapt to the changing importance of objectives over time, leading to imbalanced optimization and reduced training efficiency (Lee et al., 2025), ultimately limiting the upper bound of generation quality. In contrast, MultiTune leverages the phase-aware progression of diffusion processes. It performs temporal decoupling to enable ordered switching between objectives and adaptively adjusts the balance between phase-specific and global objectives based on noise dynamics. In summary, our approach differs from existing multi-objective optimization methods in the following aspects:*

- *(1) Fully Decoupled Training, No Fusion Needed: Existing multi-objective frameworks in this field mainly fall into two categories: reward fusion and reward soup. Reward fusion requires linearly combining multiple objective rewards with fixed weighting coefficients, which often introduces severe objective conflicts during joint optimization. Reward soup, on the other hand, trains separate models for each objective and later merges their parameters, yet the final weight fusion still cannot fully avoid conflicts arising from mixing heterogeneous objectives. In contrast, our method leverages the phase-wise generation nature of diffusion models, assigning each objective to the phase where it is most relevant and optimizing them independently. Because these phases follow a natural sequential order, no post-hoc decoupling or model fusion is required. This design maximally eliminates cross-objective interference and prevents the conflicts inherent in fusion-based approaches.*

- *(2) No Fusion Coefficients Required Across Objectives: Existing mainstream multi-objective paradigms mentioned in (1) both involve fusion operations across different objectives, and therefore require setting fusion ratio coefficients between them. These coefficients are extremely difficult to determine, because the gradient scales and sensitivities of different objectives vary significantly, and even slight deviations can lead to unstable optimization directions. In our approach, the phase objectives are temporally decoupled, so no fusion is required, and consequently no fusion ratio coefficients are needed.*

#### A.4.2 DIFFERENCES FROM EXISTING RL-BASED DIFFUSION FINE-TUNING METHODS

- （1）*Phase-Wise Multi-Objective Optimization vs. Full-Process Single-Objective Optimization: Unlike existing RL fine-tuning methods that generally adopt a paradigm of optimizing a single objective throughout the entire process, MultiTuneleverages the stage-wise generation mechanism of diffusion models from structural formation to textural refinement to decouple the multi-objective optimization process along the temporal dimension. Traditional methods continuously reinforce the same task objective across the entire denoising trajectory, causing the model to receive identical optimization signals in both early and late stages of training. This not only fails to accommodate the differences in semantic sensitivity across stages but also easily leads to quality bias caused by over-optimizing a single objective. In contrast, MultiTuneintroduces the most appropriate objectives into the structural, textural, and marginal phases, respectively, ensuring that each objective is optimized during the stage where it contributes most. This design achieves intra-phase focus and inter-phase orderly coordination, fundamentally avoiding competition and interference among objectives. Such a phase-wise design provides a natural structural foundation for multi-objective collaboration and constitutes a key distinction from existing single-objective fine-tuning methods.*

- （2）*Adaptive Compute Optimization vs. No Compute Optimization: Unlike existing RL fine-tuning methods for diffusion models that typically do not explicitly consider computational cost, MultiTuneincorporates an adaptive computation optimization mechanism from the outset of its design. Traditional methods perform full-length sampling and full reward computation along the entire denoising trajectory, resulting in a large amount of redundant*

*computation that cannot be skipped and thus leading to high training costs and low resource utilization efficiency. In contrast, MultiTunedetects the convergence trend of the texture reward at the end of each phase, enabling adaptive early termination of redundant steps, and dynamically filters out trajectories that provide no benefit—or even have negative effects—on optimization at the sample level. This computation-aware mechanism significantly reduces ineffective computation, allowing the model to complete training at a lower cost while maintaining performance, thereby forming a fundamental distinction from methods without computation optimization.*

- （3）**Explicit Exploration Incentives vs. Absence of Exploration Mechanisms** *Diffusion models typically involve long denoising sequences, and the local decision at each step influences the final generated result. In such high-dimensional and long-horizon generation tasks, traditional RL fine-tuning methods without explicit exploration mechanisms are prone to accumulating early errors during the mid- and late stages and quickly collapsing into a single trajectory, thereby losing the ability to search for potentially better directions. In contrast, MultiTuneintroduces explicit exploration incentives to encourage appropriate structural perturbations at each stage, enabling high-level semantic layout and mid-level textural structure to be optimized simultaneously. Explicit exploration not only improves the searchability of the denoising process but also significantly enhances the diversity and quality of optimization during training, forming a decisive distinction from methods without exploration mechanisms.*

- **Differentiated Mid-Step Rewards vs. Non-Differentiated Mid-Step:** *In long-horizon generation tasks, non-differentiated intermediate-step rewards often exhibit low information density and weak guidance, and may even introduce noisy gradients at certain stages, thereby undermining the stability and efficiency of RL fine-tuning. Because such reward shaping does not distinguish between the structural construction in the early denoising steps and the texture refinement in the later steps, it often fails to provide effective learning signals for the model. In contrast, MultiTuneassigns differentiated rewards that correspond to the generative characteristics of each stage, such that early-stage rewards focus more on structural consistency, mid-stage rewards emphasize semantic alignment, and late-stage rewards target detail enhancement and noise suppression. This differentiated intermediate-step reward design substantially increases the effective information provided at each step and reduces useless or even misleading gradients during training, constituting a key advantage over methods with non-differentiated rewards. Rewards*

- （5）**Denoising-Aware Optimization vs. Non–Denoising-Aware Optimization:** *Existing RL fine-tuning methods typically treat the diffusion process as a homogeneous sequence, applying a uniform optimization strategy to all denoising steps without explicitly modeling the semantic and structural differences across stages. Such non–denoising-aware optimization overlooks the stage-wise progression of diffusion models—from high-level structural formation to detailed refinement—causing early and late steps to receive homogeneous gradient signals. This not only weakens training effectiveness but also easily introduces cross-stage gradient interference. In contrast, MultiTuneadopts a denoising-aware optimization strategy that identifies the generative characteristics of different denoising stages and applies the most appropriate optimization signals to structural construction, semantic shaping, and texture refinement, respectively. This alignment between the optimization process and the inherent diffusion dynamics significantly improves generation quality and training stability. Such a stage-aware optimization design constitutes the core distinction from non–denoising-aware methods.*

- （6）**Exploration–Exploitation Balance Methods vs. Methods Without Exploration–Exploitation Modeling.** *In the long-horizon decision-making setting of diffusion models, the exploration–exploitation balance is particularly crucial, as early decisions exert a cascading influence on subsequent steps. However, methods that do not model the exploration–exploitation balance often rely on deterministic or weakly stochastic policy updates, causing the model to fall into a fixed generation trajectory early in training and thereby miss the possibility of discovering better structural or semantic patterns. In contrast, MultiTuneadopts an exploration–exploitation–aware strategy that explicitly encourages the model to attempt diverse structural variations at different stages, enabling the optimization process to escape local optima and maintain active search within the high-dimensional policy space, thus improving generation quality and diversity. This explicit modeling of*

*exploration constitutes the fundamental distinction from methods that do not incorporate exploration–exploitation modeling.*

## B   RL in Diffusion Models

We formulate the denoising generation process of diffusion models as a Markov Decision Process (MDP), represented as a 6-tuple: $\langle \mathcal{S}, \mathcal{A}, P, \rho_0, R, T \rangle$, where: $\mathcal{S}$ denotes the state space, with each state corresponding to the current intermediate image representation; $\mathcal{A}$ is the action space, representing the predicted noise at each step; $P$ is the state transition function, conforming to the physical formulation of the diffusion process; $\rho_0$ is the initial state distribution, typically a standard Gaussian noise; $R$ denotes the reward function, representing feedback on generation quality; and $T$ is the total number of denoising steps, corresponding to the time horizon of the diffusion model. Formally, the MDP is defined as:

$$\mathbf{s}_t \triangleq (\mathbf{z}, t, \mathbf{x}_{T-t}) \in \mathcal{S}, \mathbf{a}_t \triangleq \mathbf{x}_{T-t-1},$$

$$\pi_\theta (\mathbf{a}_t \mid \mathbf{s}_t) \triangleq p_\theta (\mathbf{x}_{T-t-1} \mid \mathbf{x}_{T-t}, \mathbf{z}),$$

$$P (\mathbf{s}_{t+1} \mid \mathbf{s}_t, \mathbf{a}_t) \triangleq (\delta_{\mathbf{z}}, \delta_{t+1}, \delta_{\mathbf{x}_{T-t-1}}),$$

$$\rho_0 (\mathbf{s}_0) \triangleq (p(\mathbf{z}), \delta_T, \mathcal{N}(\mathbf{0}, \mathbf{I})),$$

$$R (\mathbf{s}_t, \mathbf{a}_t) \triangleq \begin{cases} r (\mathbf{x}_0, \mathbf{z}) & \text{if } t = T - 1, \\ 0 & \text{otherwise.} \end{cases}$$

where $\delta(\cdot)$ is the delta measure, and $\mathbf{z}$ is the condition. Based on this formulation, the objective of RL fine-tuning is to maximize the expected cumulative reward:

$$\mathcal{J}(\theta) = \mathbb{E}_{\pi_\theta} \left[ \sum_{t=1}^{T-1} R (\mathbf{s}_t, \mathbf{a}_t) \right],$$

Here, $\pi_\theta$ denotes a policy parameterized by $\theta$, which outputs an action at each step based on the current state $\mathbf{s}_t \in \mathcal{S}$, the timestep $t$, and the conditional input $\mathbf{y}$: $\pi_\theta(\mathbf{a}_t \mid \mathbf{s}_t, t, \mathbf{y})$ Policy optimization typically employs REINFORCE-style gradients:

$$\nabla_\theta \mathcal{J}(\theta) = \mathbb{E}_{\pi_\theta} \left[ \sum_{t=1}^{T-1} R (\mathbf{s}_t, \mathbf{a}_t) \cdot \nabla_\theta \log \pi_\theta(\mathbf{a}_t \mid \mathbf{s}_t, t, \mathbf{y}) \right].$$

## C   Phase and Main objectives

This paper categorizes the optimization objectives into two types: Phase Objectives ($O^{phase}$) and Main Objective ($O^{main}$). The former dynamically adjusts the optimization focus in line with the denoising process, which progressively refines from global structure to local details. The latter corresponds to specific downstream requirements, such as aesthetics, compressibility, and incompressibility. The two types of objectives establish a hierarchical system that combines local guidance with global convergence, as detailed below:

- Phase Objectives: During the intermediate denoising phase of diffusion, considering that structure generation is primarily determined in the early stages (Hu et al., 2025; Xie & Gong, 2025), we introduce a reward function (e.g., CLIP) assessing text-image alignment for constrained optimization. Later, as the model focuses on texture and local details, we transition to optimizing reward functions considering texture and aesthetic preferences (e.g., PickScore). Note that the framework is extensible to other objectives; see Section 3.4 for ablation studies.

- Main Objective: These objectives reflect the main requirements of specific downstream tasks. They are typically evaluated after generation completion and direct model optimization toward the main directions.

This design enables the decoupling and independent scheduling of phase-specific objectives and their adaptive fusion with the Main objective, collaboratively enhancing comprehensive quality across multiple dimensions. See Alg. 1 for the overall algorithm of our method.

---

**Algorithm 1:** Algorithm of MultiTune

**Input:** Prompt set: $\mathcal{S}$; Training epoch: $\mathcal{E}$; Denoising step: $\mathcal{T}$.

1   Initialize pretrained diffusion model $\epsilon_\theta$;

2   **for** $e = 1$ **to** $\mathcal{E}$ **do**

3     **for** *Prompt $s$ in $\mathcal{S}$* **do**

4       generate sample trajectory of $s$ iteratively

5       $\{\mathbf{x}^s_{\mathcal{T}-1}, ..., \mathbf{x}^p_0\} = \{\mu(\mathbf{x}^p_\mathcal{T}, t) + \sigma_\mathcal{T}\mathbf{z}, ..., \mu(\mathbf{x}^p_1, 1) + \sigma_1\mathbf{z}\}$

6       compute extrinsic reward

7       $R_{ext} = r(\mathbf{x}^p_0, \mathbf{z})$

8       initialize $O^{phase}$ function

9       $F = F^{str}$

10      **for** *Timestep $t$ in reversed $\mathcal{T}$* **do**

11        perform one step of denoising

12        $\mathbf{x}^p_t = \mu(\mathbf{x}^p_{t+1}, t) + \sigma_t\mathbf{z}$

13        compute intrinsic reward

14        $R^{(t-1)}_{int} = \|\hat{x}_0(x_t) - \hat{x}_0(x_{t+1})\|^2_2$

15        compute variation of $O^{phase}$

16        $\Delta O^{phase}_t = F(\hat{x}_0(x_t)) - F(\hat{x}_0(x_{t+1}))$

17        compute phase-switch indicator

18        $\mathbb{I} = (\Delta O^{phase}_t \to 0) \& (\Delta^2 O^{phase}_t \to 0)$

19        **if** $\mathbb{I} = 1$ **then**

20          **if** $F = F^{str}$ **then**

21            $F = F^{tex}$;

22          **else**

23            **break**;

24       Filtering negative $\Delta O^{phase}_t$

25       $\Delta O^{phase}_t = Filter(\Delta O^{phase}_t)$

26       optimizing final reward via PPO

27       $R_t = \Delta O^{phase}_t \cdot R^{int}_t + R^{ext}_t$

**Output:** learned model parameter $\theta$.

---

# D   MORE EXPERIMENTS

## D.1   HYPER-PARAMETERS AND IMPLEMENTATIONS

We train our model with Adam optimizer (Kingma, 2014) and a learning rate of $3 \times 10^{-4}$. The full list of hyper-parameters in our paper is shown in Table 6. For each method and each RL objective, we ran five different seeds and report the mean and standard deviation of reward on 64 randomly sampled prompts as validation set. We train each model with a total number of 25000 samples. Each experiment is conducted on a single machine with 8 NVIDIA A100 GPUs. Following (Black et al., 2023), we use the LAION aesthetics predictor for conducting the aesthetics experiments. The weight of combining multi-objective for MORL and Soup is set to $AES : CLIP : PS = 6 : 2 : 2$. The particle number used for DAS is set to 4. To achieve $a \to b$, we directly deploy `np.isclose(a,b)`, which can calculate the approximation with default $\epsilon = 1 \times 10^{-5}$.

## D.2   ALTERNATIVES FOR STRUCTURAL OBJECTIVE

*Considering the first phase in MultiTune is also not fixed as long as the objective is measuring the structural difference, we introduce two alternatives to replace the original CLIP reward model. To be specific, we deploy LAION-CLIP Schuhmann et al. (2022b) and ALIGN Jia et al. (2021) to replace the original CLIP, respectively. As shown in Fig. 13, we present the results on the Phase texture (Phase*

Table 6: Hyper-parameters in our experiment.

| Name | Description | Value |
|------|-------------|-------|
| $lr$ | learning rate of MultiTune | 3e-4 |
| optimizer | type of optimizer | Adam |
| $\xi$ | weight decay of optimizer | 1e-4 |
| $\epsilon_{gcn}$ | Gradient clip norm | 1.0 |
| $\beta_1$ | $\beta_1$ of Adam | 0.9 |
| $\beta_2$ | $\beta_2$ of Adam | 0.999 |
| $T$ | total timesteps of inference | 50 |
| $bs$ | train batch size per GPU | 2 |
| $bs_{sample}$ | sample batch size per GPU | 8 |
| $n$ | number of batch samples per epoch | 4 |
| $\eta$ | eta parameter for the DDIM sampler | 1.0 |
| $G$ | gradient accumulation steps | 4 |
| $w$ | classifier-free guidance weight | 5.0 |
| $\epsilon_{zero}$ | threshold used to define $\Delta \to 0$ | 0.002 |
| $mp$ | mixed precision | fp16 |

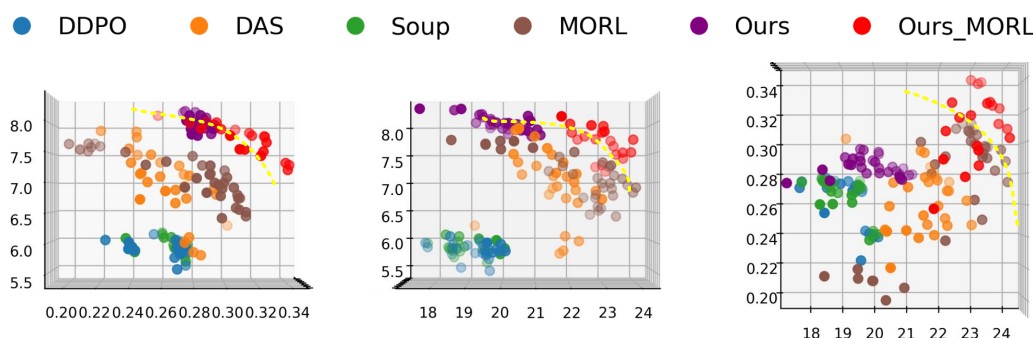

Figure 12: Three-view diagram of Fig. 3, representing the pairwise 2D Pareto front for the three objectives. The yellow dotted line denotes the optimal curves in the figures.

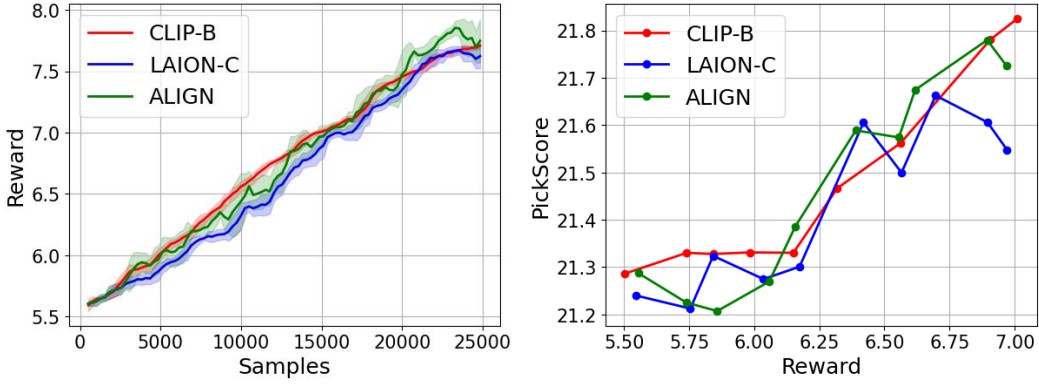

Figure 13: *Results of image-prompt alignment alternatives.*

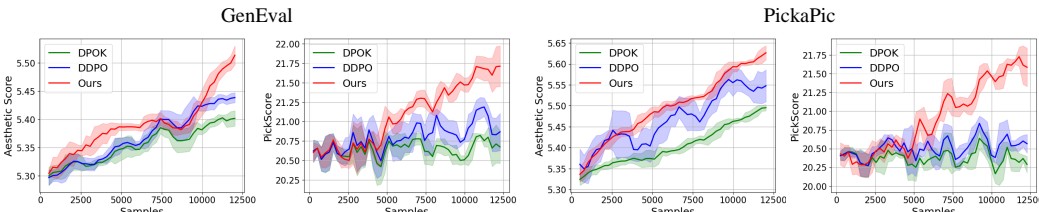

Figure 14: *Multi-objective optimization results on GenEval and PickaPic.*

*2) and the main objective. We do not present the results on the Phase structure because different alignment objectives are not consistent for measurement. It can be observed that different alignment models exhibit similar effectiveness, indicating the robustness and scalability of our method.*

### D.3 FURTHER TRAINING DATASET

*Besides the rather easy simple-animal dataset, we introduce two advanced prompt set for training, i.e., PickaPic Kirstain et al. (2023) and GenEval Ghosh et al. (2023). For baselines, we reproduce DDPO and DPOK in a MORL manner. Then, we show the curve of learning textural objective (PS) and the main objective (AES) on these datasets. As shown in Fig. 14, learning on complex prompt set is relatively harder than the simple one, hence the learning rate is also reduced. Still, our method achieves the best performance among the baselines across all metrics.*

### D.4 2-D PARETO OPTIMAL FRONT

As shown in Fig. 12, we present the three-view diagram of Fig. 3, which corresponds to the pairwise 2D Pareto front for the three objectives. For ease of comparison with Fig. 3, we directly obtained the three views by rotating the 3-D figure. It is clearly observable that our method consistently achieves the frontier performance in each objective group.

### D.5 DETAILS OF SUBJECTIVE EVALUATIONS

Here, we first provide the instructions that we deployed to guide the judgment of ChatGPT-4o:

- **Semantic Alignment**: Does the generated image accurately and comprehensively convey the key entities, scenes, and actions described in the text? (Measures the degree of direct correspondence between the textual content and the visual output.)

- **Structural Faithfulness**: Does the image exhibit any structural anomalies, such as disproportionate elements, extra limbs, or misaligned backgrounds? (Assesses the logical coherence and structural plausibility of the visual composition.)

- **Aesthetic Appeal**: Which image demonstrates superior appeal in terms of visual style, composition, color harmony, and overall aesthetics? (Reflects traditional notions of visual attractiveness.)

- **Fine-grained Detail**: Which image exhibits greater finesse and naturalism in rendering textures, materials, shadows, and other fine-grained visual details? (Reflects perceptual resolution and detail fidelity.)

- **Prompt Responsiveness**: Which image more accurately reflects the attribute constraints specified in the prompt, such as color, quantity, or action? (Used to evaluate the controllability and precision of prompt adherence.)

Then, Fig. 16 illustrates the user interface and scoring scheme designed for human evaluation. This interface supports a clearer understanding of how human ratings were collected in our study. Five evaluation dimensions are presented, the same as for LVM: Structural Faithfulness, Aesthetic Appeal, Fine-grained Detail, Semantic Alignment, and Prompt Responsiveness. Participants assign scores from 1 to 10 using slider bars, and the system automatically computes the overall average score. This design encourages consistent and multifaceted human assessments to ensure data reliability. Using

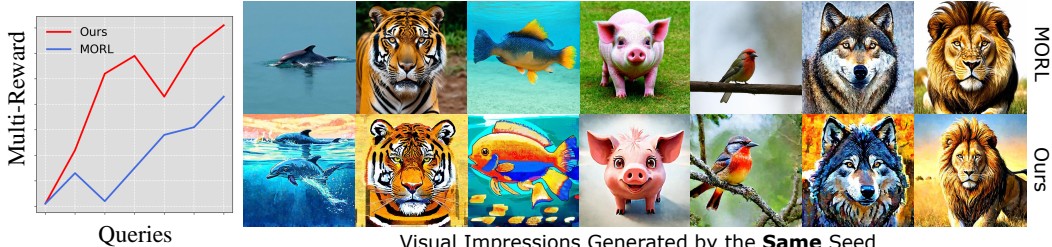

Figure 15: *Optimization results on SD3.*

this interface, a total of 1296*5 valid data points from different methods were collected, and the mean values were computed for analysis and presentation.

Table 7: *Comparisons for multi-objective optimization across various metrics on SD3.*

| Method | AES | PS | IR | CLIP | LPIPS | IS | BRISQUE$^{\downarrow}$ |
|--------|-----|-----|-----|------|-------|-----|---------|
| SD3 | 5.671 | 21.69 | 0.8541 | **0.236** | **0.664** | **23.51** | 12.52 |
| DDPO | 5.952 | 22.36 | 0.8105 | 0.233 | 0.599 | 21.91 | 11.35 |
| Ours | **6.137** | **22.80** | **0.8971** | 0.234 | 0.646 | 23.17 | **9.98** |

### D.6    RESULTS ON THE ADVANCED SD3

*To further demonstrate the applicability of the proposed MultiTune, we conduct experiments on the transformer-based Stable Diffusion v3 Liu et al. (2025). Specifically, we compare our method with MORL and the baseline with PickScore+AestheticScore as the multi-objective, simple animal as the prompt set. As shown in Fig. 15, we achieve superior optimization efficiency and thus generate images with improved aesthetic style based on the same seeds. Moreover, as shown in Tab. 7, our method exhibits superior overall generative quality while maintaining the alignment and diversity of the original model.*

## E    SUPPLEMENTARY FOR PHASE SWITCHING

*We have the following three equations to decide phase switching:*

$$t_{end}^{Phase1} = \min \left\{ t \mid \left| \Delta O_t^{Phase1} \right| \to 0 \ \wedge \ \left| \Delta^2 O_t^{Phase1} \right| \to 0 \right\}. \tag{11}$$

$$t_{end}^{Phase2} = \min \left\{ t \mid \left| \Delta O_t^{Phase2} \right| \to 0 \ \wedge \ \left| \Delta^2 O_t^{Phase2} \right| \to 0 \right\}. \tag{12}$$

$$t_{start}^{Phase3} = t_{end}^{Phase2} = \min \left\{ t \mid I_{tran(Phase1)} = 1 \ \wedge \ I_{tran(Phase2)} = 1 \right\}. \tag{13}$$

## F    DECLARATION OF LLM USAGES.

We partially employed an LLM to enhance the clarity of our English expression. However, the formulation of ideas, theoretical proofs, and experimental work were conducted entirely independently, without any LLM involvement.

## G  EXPERIMENTAL LIST

*This paper demonstrates the stability and broad effectiveness of MultiTunethrough a large number of experiments (**21 in total**), with the detailed list provided below:*

- *(1). **Motivation Justification.** Fig. 1: Visual experiments supporting the design motivation.*
- *(2). **3D Pareto Representation.** Fig. 3: Comparison with multiple multi-objective optimization methods and RL-fine-tuned diffusion models.*
- *(3). **Backbone Replacement Experiment.** Tab. 1: Demonstrates that changing the backbone does not affect the superiority of the proposed method.*
- *(4). **Cross-Reward Generalization Experiment.** Fig. 4: Verifies generalization across different reward metrics.*
- *(5). **Comparison with Single-Objective SOTA.** Fig. 5: Direct comparison with state-of-the-art single-objective methods.*
- *(6). **Computational Cost Comparisons.** Fig. 6: Runtime comparison under the same target score.*
- *(7). **Computational Overhead per Component.** Tab. 2: Time and memory consumption for each proposed component.*
- *(8). **Stage Goal Swap Experiment.** Fig. 7 (Phase 2 and Main) and Fig. 13 (Phase 1): Validates the flexibility of swapping stage objectives.*
- *(9). **Ablation Study.** Tab. 2: Ablation results of different modules.*
- *(10). **Visualization of Trajectory Diversity.** Fig. 8: Demonstrates that intrinsic rewards encourage exploration.*
- *(11). **Generalization on Complex Prompts.** Fig. 9: Performance on complex linguistic prompts.*
- *(12). **Alignment Validation.** Fig. 11: Verification on alignment requirements such as color and object count.*
- *(13). **Diversity in Generated Results.** Fig. 11: Visualization of generation diversity.*
- *(14). **Human Evaluation.** Fig. 10: Human preference study.*
- *(15). **Training on Complex Datasets.** Tab. 4: Performance on more complex datasets during training.*
- *(16). **Inference on Complex Datasets.** Tab. 3: Results of inference-time evaluation on complex datasets.*
- *(17). **Advanced Backbone Evaluation.** Tab. 5: Results on larger backbones (e.g., SDXL).*
- *(18). **2D Pareto Front Experiment.** Fig. 12: Two-dimensional Pareto analysis.*
- *(19). **Additional Datasets: GenEval and Pick-a-Pic.** Fig. 14: Evaluation on additional benchmarks.*
- *(20). **Fine-tuning on SD3 (DiT-based Model).** Fig. 15 and Tab. 7: Performance on the latest SD3 architecture.*
- *(21). **Results on the Base Backbone.** Tab. 5: Performance without any design.*

# Human preference investigation

Welcome to this image quality evaluation study. The goal of this experiment is to collect your subjective ratings of AI-generated images across multiple quality dimensions. Please use the slider bars (ranging from 1 to 10) to rate each image based on the following four dimensions:

- **Structural Faithfulness (SF)** : Does the image exhibit realistic and coherent spatial structure and composition?
- **Aesthetic Appeal (AA)** : How visually pleasing is the image in terms of style, color, and overall design?
- **Fine-grained Detail (FD)** : Does the image contain clear, rich, and naturally rendered fine details?
- **Semantic Alignment (SA)**: How well does the image content align with the intended prompt or semantic meaning?
- **Prompt Responsiveness (PR)**: How accurately and completely does the image reflect the specific content or instructions described in the prompt?

After you complete the ratings for all five dimensions, the system will automatically calculate and display the overall average score for the image. Please provide thoughtful and honest evaluations to ensure the reliability of the collected data. Thank you for your participation!

Prompt : Four roses

| | |
|---|---|
| Structural Faithfulness | 8 |
| Aesthetic Appeal | 9 |
| Fine-grained Detail | 7 |
| Semantic Alignment | 9 |
| Prompt Responsiveness | 9 |
| Overall Score | 8.4 |

Previous          Next

Figure 16: Interface Design and Scoring Details for Subjective Evaluation

