# OpenReview forum: "MultiTune: Phase-Aware Multi-Objective Optimization for Diffusion Models"
_ICLR.cc/2026/Conference — Submitted to ICLR 2026_

### Official Review · Reviewer_zFFf · 2025-10-30

**Soundness:** 3
**Presentation:** 3
**Contribution:** 3
**Rating:** 6
**Confidence:** 4

**Summary:**

The paper proposes a novel multi-objective RL framework for text-to-image diffusion models by introducing an additional phase objective that gradually changes throughout the generation process. The proposed phase objective serve to mitigate the sparse reward problem, where the main reward can only be evaluated on the fully generated image. The authors conducted extensive experiments and demonstrated that the proposed method outperforms existing RL baselines.

**Strengths:**

1. The presentation is clear and easy to understand. The Phase reward is well-motivated and different phases align with intuitive understanding about the diffusion generation process.
2. The experiments are comprehensive, incorporating a wide range of metrics (CLIP, Pickscore) and base models (SDv1.5, SD2), demonstrating that the proposed method is universally applicable. The author also provided subjective evaluations using human and VLM judge, further strengthening their results.
3. The author provided a thorough ablation studies on various design choices.

**Weaknesses:**

1. The author experimented mostly on SD-series U-Net, it is unclear if the proposed method works for more recent DiT models based on rectified flow formulation, such as SD3, Sana, Flux, etc.
2.  Given 1, the result in table 5 is especially concerning, as it raises the issue of the scalability of the proposed method. While SD-XL is a stronger base model, the performance after RL fine-tuning is actually worse than SDv1.4. The author should include a row of base model's performance in this table so it is easier to see the improvements. I imagine the improvements with respect to base model will be smaller? If that is the case, the author should provide additional discussion.

**Questions:**

1. What  exactly are the phase reward models? In Appendix C, the author used ambiguous terms like " introduce a reward function (e.g., CLIP)", "considering texture and aesthetic preferences (e.g., PickScore)". Is the reward model just CLIP and PickScore? Are other models also used? The authors should provide more concrete details for better reproducibility
2. What is "see Section X.X " in page 15 L 806
3. Figure 13 on Page 19 appears to be broken in PDF

---

> ### Author Response · Authors · 2025-11-18
> **Response to Reviewer zFFf**
>
> Thank you very much for your positive evaluation and valuable comments. We have carefully addressed and clarified all the issues you suggested. The revised PDF has been updated, with all changes highlighted in italic blue for your reference. Here, we provide point-by-point responses.
>
> ---
>
> ## Weakness 1
> > The author experimented mostly on SD-series U-Net, it is unclear if the proposed method works for more recent DiT models based on rectified flow formulation, such as SD3, Sana, Flux, etc.
>
> Thank you for your valuable suggestion. **To the best of our knowledge**, no existing work on RL fine-tuning of SD has considered a more advanced approach than **SDXL**. Furthermore, due to the **fundamental architectural differences**, adapting the method would be extremely challenging. However, we do agree that the issue raised by the reviewer is **crucial**. Therefore, we are currently working on **reproducing the method for SD3**, and due to **time constraints**, we will provide the results as soon as possible.
>
> ---
>
> ## Weakness 2
> > Given 1, the result in table 5 is especially concerning, as it raises the issue of the scalability of the proposed method. While SD-XL is a stronger base model, the performance after RL fine-tuning is actually worse than SDv1.4. The author should include a row of base model's performance in this table so it is easier to see the improvements. I imagine the improvements with respect to base model will be smaller? If that is the case, the author should provide additional discussion.
>
> Thank you for your question. As requested, we first present the baseline performance of the models (SD14, SDXL) in **Table 5**. It can be seen that the baseline performance of **SDXL** is **overall superior** to that of **SD14**. This indicates that in subsequent RL fine-tuning, **SD14** is more easily fitted to the reward function, achieving higher scores. This might be because SDXL has a **much larger number of parameters and training data scale**, hence greater robustness compared to SD14, making it less susceptible to external signals, which results in a much slower fitting of the reward function. In fact, we observed that after increasing the **batch size** and extending the **training iteration time**, SDXL can also achieve higher reward scores. This further demonstrates that SDXL indeed has a **slower and more cautious learning efficiency**, which is closely related to its larger parameters and pretraining data scale. We will add this discussion to our manuscript.
>
> ---
>
> ## Question 1
> > What exactly are the phase reward models? In Appendix C, the author used ambiguous terms like " introduce a reward function (e.g., CLIP)", "considering texture and aesthetic preferences (e.g., PickScore)". Is the reward model just CLIP and PickScore? Are other models also used? The authors should provide more concrete details for better reproducibility.
>
> Thank you for your question. Regarding the reward function, we are **not limited to any specific model version**. **Specifically**, for the **structural phase (Phase 1)**, we use **CLIP** because its scoring is entirely based on **image-text alignment**, without unnecessary consideration of aesthetics, human preferences, or other factors that are not relevant to Phase 1. In **Appendix D.2**, in addition to the current **ViT-B/32**, we also provide results where CLIP is replaced with the semantic alignment models **LAION-C** and **ALIGN**. As seen, there is **no significant difference** in the results, which demonstrates the **robustness and scalability** of our method for aligning reward functions.
>
> For the **Textural Phase (Phase 2)**, we have already tried replacing **PickScore** with **IR** in **Figure 7** of the paper, which shows that our second phase can be successfully extended to other **human preference models**.
>
> ---
>
> ## Question 2
> > What is "see Section X.X " in page 15 L 806
>
> We sincerely apologize for this oversight. In fact, we were referring to **Section 3.4**, and we have made the corresponding corrections.
>
> ---
>
> ## Question 3
> > Figure 13 on Page 19 appears to be broken in PDF
>
> Thanks for your comment. This may be due to a **compatibility issue with the PDF**. You can try viewing the PDF directly in **Chrome** by clicking the PDF file in the top-right corner of the submission interface. We have verified that the PDF is intact in this case, and we will do our best to resolve the issue.

---

> ### Comment · Reviewer_zFFf · 2025-11-19
>
> Hi
> Thanks to the authors for the response. I have a couple of feedbacks.
>
> **No existing work on RL fine-tuning of SD has considered a more advanced approach than SDXL.**
>
> This is categorically not True.  I am aware that much old literature do not use more advanced models than SDXL, however many works this year such as Flow-GRPO (8 May 2025, using SD3, now has 70+ citations) have moved towards DiT architecture. I hope the authors properly acknowledge these recent works. Additionally, I am fully aware that training SD3 requires considerable compute and may not feasible for the rebuttal period. To clarify, I'm not specifically expecting any large-scale experiments at the rebuttal period, I'm just stating my position failing to show the proposed method works for more recent architecture like Flux or SD3 will relatively limit the impact of the proposed method (comparing with other recent works), and preventing me from giving a higher score. I do not see this omission as grounds for rejection, though.
>
> **SDXL training is slower and more cautious learning efficiency**
>
> I think it is fine if "Increasing the batch size and extending the training iteration time" is necessary to obtain higher scores, since even SFT of larger models also requires more training time. Is the results of larger BS and longer training included anywhere? At the moment, SD-XL+proposed method in Table 5 not only has lower reward scors than SDv1.4, but even marginally lower CLIP score than baseline in Table 5.
>
> In general, it is fine if larger models need more training to reach comparable or better performance of SD v1.4, but I would question the utility of this work if "after more training, SDXL still cannot reach or outperfom SDv1.4". Given this, I suggest the authors to elaborate on additional experiments with larger BS/longer training.
>
> [1] Flow-GRPO: Training Flow Matching Models via Online RL

---

> > ### Author Response · Authors · 2025-11-19
> >
> > We sincerely appreciate your insightful and constructive suggestions. Following your latest recommendations, we will provide more in-depth discussion and conduct further experiments. Thank you for your continued support — we will work promptly to fulfill your new requirements. We are deeply grateful for the time and care you have devoted to reviewing our manuscript and for the excellent advice you have provided.

---

> > ### Author Response · Authors · 2025-11-22
> > **Response to Esteemed Reviewer zFFf with followup SD3 results.**
> >
> > Thank you very much for your valuable feedback and patience. Based on your careful guidance, we are pleased to report that, through our continued efforts, we have now completed experiments on the more advanced DiT-based SD3 model (`stabilityai/stable-diffusion-3-medium-diffusers`). **Concisely, our method continues to achieve superior multi-objective optimization performance.**
> >
> > Specifically, we added **Appendix D.6** in the revised PDF. As shown in **Fig. 15**, our method demonstrates significantly improved optimization efficiency, and the generated images are noticeably more visually appealing under the **same seed**. Furthermore, as presented in **Table 7**, our approach achieves stronger overall aesthetic improvement while better preserving the original model’s alignment and diversity.
> >
> > We sincerely hope that these enhanced results further meet your high expectations and may, as you mentioned, lead you to consider a higher evaluation score.

---

> > ### Author Response · Authors · 2025-11-26
> >
> > Dear Reviewer,
> >
> > We hope this message finds you well. **We have completed the additional SD3 experiments as you requested**. As the rebuttal deadline is approaching, we would be deeply grateful if you could spare a moment from your busy schedule to take a look at our updated responses and results.
> >
> > Thank you sincerely for your time and consideration.

---

> > > ### Comment · Reviewer_zFFf · 2025-11-26
> > >
> > > Hi
> > >
> > > Thanks authors for the additional work and discussions. I'm more convinced of the soundness and impact of this work and decided to raise my score. I strongly suggest the authors move SD3 results to the main paper in their camera-ready version, as it would significantly boost the perceived impact of this work.

---

> > > > ### Author Response · Authors · 2025-11-26
> > > >
> > > > Thank you for your recognition of our work. On behalf of all the authors, I would like to express our sincere respect to you.

---

> > > > ### Author Response · Authors · 2025-11-26
> > > >
> > > > We greatly appreciate your insightful suggestion to fine-tune SD3 using our method. Your recommendation is highly valuable to us and will undoubtedly help improve our work. Thank you sincerely for your constructive and thoughtful comments.

---

> ### Author Response · Authors · 2025-11-23
> **Experiment List Specially Compiled for the Esteemed Reviewer zFFf**
>
> Dear Esteemed Reviewer zFFf,
>
> Thank you very much for your thoughtful evaluation. Our paper provides a substantial number of experiments to support assessment by the reviewers. To facilitate your comprehensive understanding of the experimental evidence, we have compiled a complete list of all experiments conducted in this work.
>
> **In total, the paper includes 20 different types of experiments**. For your convenience, we have marked in italics the experiments that may be of particular interest to you based on your review comments, allowing you to locate them directly and efficiently.
>
> We believe the experimental validation in our paper is comprehensive and exceeds that of many existing works.  which include:
> 1. Motivation Justification: Fig. 1 (visual experiments).
> 2. 3D Pareto Representation: Fig. 3 (comparison with multiple multi-objective optimization methods and RL fine-tuned diffusion models).
> 3. Backbone Replacement Experiment: Tab. 1 (shows backbone replacement does not affect method superiority).
> 4. ***Cross-Reward Generalization Experiment**: Fig. 4 (proves the generalization capability across different reward metrics)*.
> 5. Comparison with Single-Objective SOTA: Fig. 5 (direct comparison with SOTA single-objective methods).
> 6. Computational Power Experiment: Fig. 6 (time overhead for achieving the same score).
> 7. ***Stage Goal Swap Experiment**: Fig. 7 (verifying the flexibility of swapping stage objectives)*.
> 8. Ablation Study: Tab. 2 (ablation study results).
> 9. Visualization of Diversity in Generated Trajectories: Fig. 8 (exploration encouraged by internal reward).
> 10. Generalization on Complex Prompts: Fig. 9 (performance on complex prompts).
> 11. Alignment Validation: Fig. 11 (alignment for specific requirements such as color and count).
> 12. Diversity in Generation: Fig. 11 (visualizing the diversity of generated results).
> 13. Human Evaluation Experiment: Fig. 10 (human evaluation results).
> 14. Generalization on Complex Datasets (Training): Tab. 4 (performance on more complex datasets).
> 15. Inference on Complex Datasets: Tab. 3 (inference results on complex datasets).
> 16. ***Advanced Backbone Training**: Tab. 5 (results on larger backbones like SDXL)*.
> 17. 2D Pareto Experiment: Fig. 12 (2D Pareto front).
> 18. ***Structural Metric Change Experiment**: Verification of the adaptability of structural indicators*.
> 19. GenEval and Pick-a-Pic Datasets: Fig. 14 (performance on additional datasets).
> 20.  ***Fine-tuning on SD3 Model**:  Performance on the latest SD3 model*,see Fig.15 and Tab.7.
>
> We firmly believe these extensive experiments validate the robustness and generality of our method.

---

> ### Author Response · Authors · 2025-11-24
> **Summary of responses to the weaknesses pointed out by reviewer zFFf.**
>
> **Weaknesses1**:The author experimented mostly on SD-series U-Net, it is unclear if the proposed method works for more recent DiT models based on rectified flow formulation, such as SD3, Sana, Flux, etc.
>
> Please see  **Response to Esteemed Reviewer zFFf (*reply to weakness 1*)** and **Response to Esteemed Reviewer zFFf with follow-up SD3 results**. Additional results and detailed analyses are provided in the PDF **(*Fig. 15 + Tab. 7 + Appendix D.6*)**.
>
> **Weaknesses2**:Given 1, the result in table 5 is especially concerning, as it raises the issue of the scalability of the proposed method. While SD-XL is a stronger base model, the performance after RL fine-tuning is actually worse than SDv1.4. The author should include a row of base model's performance in this table so it is easier to see the improvements. I imagine the improvements with respect to base model will be smaller? If that is the case, the author should provide additional discussion.
>
> Please see the  **response to Reviewer zFFf *regarding weakness 2*** and the discussion in ***Section 3.6 of the PDF***.

---

### Official Review · Reviewer_d9rJ · 2025-10-30

**Soundness:** 3
**Presentation:** 3
**Contribution:** 3
**Rating:** 4
**Confidence:** 4

**Summary:**

This paper addresses the challenge of fine-tuning text-to-image diffusion models to satisfy multiple objectives rather than optimizing a single reward. To this end, the authors propose MultiTune, which introduces three key components: a phase-aware switching strategy, adaptive coordination of objectives, and efficiency-aware training optimization.

**Strengths:**

1. The idea of decomposing the denoising process into intuitive phases and tying objectives to those phases is meaningful.
2. The experimental evaluation is extensive, covering multiple model backbones.

**Weaknesses:**

1. The paper employs the “Simple-animals” dataset (45 classes for training, 398 for testing). To better support claims of generality, it would be valuable to include larger and more diverse prompt sets—such as GenEval [1] or datasets spanning multiple domains.
2. More details are needed on computational cost. Specifically, what is the additional overhead (in memory and compute) introduced by the phase-aware switching and dynamic balancing mechanisms compared to simpler baselines?

**Questions:**

1. How are the structural and textural scores in Figure 1 computed? What model architectures are used for these evaluations?
2. How exactly is the denoising trajectory divided into $P_{Structural}$, $P_{Textural}$ and $P_{Marginal}$. How are the corresponding timesteps determined?
3. There are a few textual inconsistencies—for example, line 806 references “Section X.X.” Could the authors clarify which section this refers to?

---

> ### Author Response · Authors · 2025-11-18
> **Response to Reviewer d9rJ (Part 1)**
>
> Thank you so much for your comments. We have carefully revised and clarified all your suggested issues, and we have updated the revised PDF with the revision marked in italy blue.
>
> ## Weakness 1
> > The paper employs the “Simple-animals” dataset (45 classes for training, 398 for testing). To better support claims of generality, it would be valuable to include larger and more diverse prompt sets—such as GenEval [1] or datasets spanning multiple domains.
>
> Thank you very much for your valuable feedback.  Regarding the training dataset diversity:
> - **Already have**: we have provided training results on Hpsv2 and ImageNet-A, as shown in Table 4.
> - **Additionally**: as per your request, we have added experiments on GenEval and PickaPic (please refer to Appendix D3 in the updated PDF).
>
> ---
> ## Weakness 2
> > More details are needed on computational cost. Specifically, what is the additional overhead (in memory and compute) introduced by the phase-aware switching and dynamic balancing mechanisms compared to simpler baselines?
>
> Thank you very much for your question.
>
> - **Already have**: Regarding computational cost, we already provide an analysis in the main text, as shown in **Table 2 and Figure 6**. In real-time, since our method can adaptively identify the marginal gains of structure and texture metrics, we are able to truncate RL training during periods where the gain trend is not significant. As a result, our method **significantly reduces computational overhead**, and the experimental results support this claim.
>
> - **Additionally**: in Table 2, we have provided new ablation experiments on **memory usage** related to the phase-aware mechanism and adaptive balancing mechanism, along with additional analysis: The results show that introducing $\Delta F$ increases the cost by about 10GB, while skipping Phase 3 **reduces the gradient accumulation step** per sample, thus saving memory.
>
> ---
>
> ## Question 1
> > How are the structural and textural scores in Figure 1 computed? What model architectures are used for these evaluations?
>
> Thank you for your question. In Figure 1, we use CLIPScore (`ViT-B/32`) as $F^{str}$ and PickScore (`yuvalkirstain/PickScore_v1`) as $F^{tex}$ to evaluate the intermediate state during the denoising process of `SDv14`.

---

> ### Author Response · Authors · 2025-11-18
> **Response to Reviewer d9rj (Part 2)**
>
> ## Question 2
> > How exactly is the denoising trajectory divided into P_str, P_tex and P_mar.  How are the corresponding timesteps determined?
>
> Thank you very much for your question. Please allow us to clarify the issue.
>
> This work follows the inherent structural and textural feature change trends of the diffusion model, as shown in Figure 1, and combines statistical analysis of feature changes during the denoising process. We divide the denoising process into three stages:
>
> 1. **Structural Phase $P_{\text{Structural}}$**
>    - **Definition**: From the high-noise starting point \(t = T\) until the structural feature changes slow down.
>    - **Criterion**: When the change in the Phase Objective approaches 0, and the second derivative of the change also approaches 0, we determine the end of this phase:
>
>    $I_{\text{tran(Phase1)}} = \left(\Delta O^{\text{Phase1}}_t \rightarrow 0\right) \land \left(\Delta^2 O^{\text{Phase1}}_t \rightarrow 0\right).$
>
>    When both conditions are satisfied, it indicates that the optimization trend for structural metrics has reached the point of diminishing returns, and further optimization would neither yield significant gains nor justify additional computational cost. Thus, we define the end of the first phase as $t_{\text{end}}^{\text{Phase1}}$, as (refer to the updated PDF Appendix E Eq. 6 )
>
> 2. **Textural Phase $P_{\text{Textural}}$**
>    - **Definition**: Following the structural phase, this phase lasts until the texture features stabilize.
>    - **Criterion**: Similar to the structural phase, we define the end of the texture phase when the Phase Objective's change and second derivative approach zero:
>
>    $ I_{\text{tran(Phase2)}} = \left(\Delta O^{\text{Phase2}}_t \rightarrow 0\right) \land \left(\Delta^2 O^{\text{Phase2}}_t \rightarrow 0\right).$
>
>    Once both conditions are satisfied, it indicates that the optimization trend for texture metrics has reached its point of diminishing returns. We define the end of the second phase as $t_{\text{end}}^{\text{Phase2}}$, as (refer to the updated PDF Appendix E Eq. 7 )
>
> 3. **Marginal Phase ($P_{\text{Marginal}}$)**
>    - **Definition**: The final stage of the denoising process, where both structural and textural changes have saturated. Continuing RL training at this stage is meaningless and only increases computational costs (which is why our method significantly reduces computational overhead compared to the baseline), exacerbating reward hijacking issues.
>    - **Criterion**: We define the start of the third phase as the point where both the structural and textural phases are complete as (refer to the updated PDF Appendix E Eq. 8 )
>
> The timestep values for the phase divisions are not fixed in advance but are adaptively determined based on the feature change trends of different prompts under standard sampling. Specifically:
> - In Phase 1 (Structural Phase), the optimization interval is $[T, t_{\text{end}}^{\text{Phase1}}]$. During this interval, our method focuses on optimizing reward metrics that reflect semantic features.
> - In Phase 2 (Textural Phase), the optimization interval is $[t_{\text{end}}^{\text{Phase1}}, t_{\text{end}}^{\text{Phase2}}]$. During this phase, our method focuses on optimizing reward metrics that reflect image texture.
> - In Phase 3 (Skip Phase), the interval is $[t^{\text{Phase3}}_{\text{start}}, 0]$. During this phase, our method skips the training intervals where there is no significant optimization gain, saving computational resources and mitigating reward hacking.
>
> ---
>
> ## Question 3
> > There are a few textual inconsistencies—for example, line 806 references “Section X.X.” Could the authors clarify which section this refers to?
>
> We sincerely apologize for this oversight. In fact, we were referring to Section 3.4 in the main text, and we have made the corresponding corrections.

---

> ### Author Response · Authors · 2025-11-19
> **Response to the Reviewer d9rJ**
>
> Dear Esteemed Reviewer d9rJ,
>
> We sincerely appreciate your insightful questions and constructive comments. We have thoroughly addressed all the issues you raised and provided a comprehensive response for your review. In addition, we have paid special attention to your suggestion regarding evaluation on more datasets. Following your recommendation, we carried out experiments on the GenEval dataset and updated the relevant references. Should you have any additional concerns or require further clarification, please do not hesitate to inform us. We will spare no effort in addressing them.

---

> ### Author Response · Authors · 2025-11-20
>
> Dear Honorable Reviewer d9rJ,
>
> Please allow me, on behalf of all authors, to extend our sincere appreciation for your thoughtful review and the time you have devoted to evaluating our work.
>
> We have carefully prepared a detailed rebuttal addressing all the weaknesses and questions you raised in the first round. For your reference, we summarize the key points as follows:
>
> 1.Additional experiments on a newly incorporated dataset have been conducted in response to your suggestions.
>
> 2.A detailed breakdown of computational costs has been provided.
>
> 4.Further information regarding the structure and texture evaluation models has been added.
>
> We have revised the manuscript accordingly and kindly invite you to review the updated PDF.
>
> If any part of our first-round rebuttal remains unclear, or if you have new questions or comments, we would be more than happy to address them promptly.
>
> Thank you once again for your invaluable contributions to improving our work.
> Respectfully yours,
> All coauthor

---

> ### Author Response · Authors · 2025-11-21
>
> Dear Esteemed Reviewer d9rJ,
>
> We would like to express our sincere gratitude for your thorough review and for the time you have dedicated to our submission. As today marks the recommended response date for reviewers, we would be deeply grateful if you could spare a moment to share your feedback with us. We are fully confident in our ability to address all of your concerns.
>
> Thank you very much, and we wish you all the best！

---

> ### Author Response · Authors · 2025-11-22
>
> Dear Reviewers d9rJ,
>
> Thank you very much for the time and effort you have already invested in reviewing our submission. We sincerely appreciate your constructive comments, which have greatly helped us improve the quality and clarity of our work.
>
> Over the past week, we have carefully addressed all your concerns and provided substantial additional experiments, analyses, and clarifications. We fully understand that you may have very busy schedules, and we truly appreciate the workload that reviewing entails.
>
> At the same time, ICLR strongly encourages iterative discussion between authors and reviewers. With great respect for your time constraints, we kindly hope that you might take a moment to look at our recent responses when convenient. We genuinely value your feedback and would be grateful for any further thoughts you might be willing to share.
>
> Thank you again for your contributions to the reviewing process and for helping us strengthen our work.
> We truly appreciate your time and consideration.
>
> Warm regards,
> On behalf of all authors

---

> ### Author Response · Authors · 2025-11-26
>
> Dear Reviewer,
>
> As the discussion deadline is approaching, we sincerely hope that our efforts have helped address your concerns. If there are any remaining questions or points you would like to further discuss, we would be more than happy to continue the conversation and provide comprehensive clarifications at any time.

---

### Official Review · Reviewer_LokY · 2025-11-01

**Soundness:** 3
**Presentation:** 3
**Contribution:** 2
**Rating:** 4
**Confidence:** 3

**Summary:**

This paper proposes MultiTune, a reinforcement learning framework for text-to-image diffusion models that dynamically switches optimization objectives during the denoising process.
The model separates diffusion steps into three stages — structural, textural, and marginal — and adaptively balances intrinsic and extrinsic rewards.
Experiments across Stable Diffusion backbones show improvements on metrics such as AES, CLIP, and PickScore.

**Strengths:**

1. The paper focuses on the characteristics of the diffusion process  — namely, the generation proceeds from global structures to fine details — and proposes an efficient learning method for multi-objective preference optimization.

2. It conducts comprehensive experiments using multiple baselines and evaluation metrics, demonstrating performance improvements across all metrics.

**Weaknesses:**

1. It is unclear from the main text whether Equation (4) truly addresses a multi-objective task.

2. The proposed method does not guarantee functionality under arbitrary combinations of preference objectives. In particular, the structure formation phase is fixed to use CLIP as the guiding signal. Therefore, in terms of exploring the trade-offs inherent to true multi-objective preference optimization, the contribution appears somewhat limited.

**Questions:**

1. Could you tell me why CLIP and ES are used solely as feedback signals for exploration rather than being directly optimized? Equation (4) does not appear to represent multi-objective preference learning?

2. Why did the authors choose to fix the extrinsic reward to AES instead of dynamically optimizing CLIP or ES depending on the phase change?

3. Have the authors considered alternatives to using CLIP as the guiding signal during the structure formation phase?

---

> ### Author Response · Authors · 2025-11-18
> **Response to Reviewer LokY (Part1)**
>
> First, please allow me to express our sincere gratitude, on behalf of all authors, for your thorough review and the time you dedicated to it. We greatly appreciate your recognition of the soundness, presentation, motivation, and solid experiments. We understand that the reason you assigned a score of 4 instead of a higher one was due to **concerns regarding the contribution**. However, this seems to be a misunderstanding, as our **actual contribution aligns closely with what you might have expected**. Please refer to the clarification below. We have provided responses to the issues you raised in hopes of assisting you in better understanding our work.
>
> All modifications in the revised PDF are marked in italy blue.
>
> ---
> ## Weakness 1
>
> > It is unclear from the main text whether Equation (4) truly addresses a multi-objective task.
>
> The formulation of Equation (4) follows the standard reward function structure in the reinforcement learning (RL) field, which includes both external and stage rewards, as outlined in previous works [1, 2]. From this equation, it can be seen that, to address the sparse reward problem encountered when fine-tuning diffusion models with RL, our work innovatively introduces stage rewards (which include intrinsic reward components) into the RL fine-tuning paradigm. As a result, our reward function contains two parts:
>
> - **Extrinsic reward** corresponding to the traditional reward in the RL fine-tuning of diffusion models, which is the model's score at the x0 point after denoising is fully completed. This reward only appears once at the final step, and no reward is provided for earlier timesteps. This leads to the problem that, without reward signals, the optimization of earlier timesteps in the diffusion model fine-tuning process becomes highly unstable and learning is very difficult.
>
> - **Phase reward** serves as the optimization reward for the Phase Objective (supplementing Rext by providing reward signals for each previous timestep).
>
> In summary:
> The form of Equation (4) essentially adopts the standard total reward function definition used in the RL field, where the goal is not to explicitly represent the multi-objective structure but to incorporate the rewards from different stages into the RL optimization framework. The "multi-objective" of MultiTune comes from:
>
> Each stage has an independent **Phase Objective** (structure, texture), corresponding to different reward functions $R_{phase}$.
>    - **Structure Phase**: Optimizing the reward that reflects the semantic structure quality (Optimization Goal 1), guided by $R_{int}$
>    - **Texture Phase**: When the generation of the diffusion model enters the texture refinement stage, the reward shifts to optimizing the texture (Optimization Goal 2), guided by $R_{int}$
>    - **Global Preference Objective**: Throughout the entire denoising trajectory, the reward continuously optimizes user preferences (Optimization Goal 3), which is optimized directly via external reward through policy gradient optimization.
>
> Thus, the dynamic phase-switching mechanism adapts the optimization goals based on the Phase Objective. Therefore, the RL optimization throughout the entire sampling trajectory does not optimize a single objective but a **decoupled objective** composed of multiple stage-based rewards. This work optimizes three decoupled objectives: two stage-specific objectives and one main objective. It is important to note that the *diffusion model exhibits a hierarchical evolution from structure to details during the generation process*, a phenomenon that has been well discussed in previous theoretical works. Hence, **we do not consider "the staged nature of diffusion models" as the novelty** of this work but rather design a **suitable RL fine-tuning algorithm that accommodates this property**. In summary, our core contributions are:
>
> 1. Proposing a mechanism that can adaptively perceive the changes in the generation stage and reasonably embed optimization objectives matching the characteristics of each stage.
> 2. Performing multi-objective RL optimization through stage-decoupled methods.
>
> To the best of our knowledge, ***this RL method that links and decouples various preference objectives with the natural evolution process of diffusion models has not been explored in previous works.***
>
> In other words, Equation (4) does not attempt to explicitly model multi-objectives with a single formula, but rather represents the standard form used in RL frameworks to calculate the total reward. The true multi-objective structure is achieved through our phase-dependent reward definitions and dynamic phase switching.
>
> [1] Exploration by random network distillation
>
> [2] Never give up: Learning directed exploration strategies

---

> ### Author Response · Authors · 2025-11-18
> **Response to Reviewer LokY (Part 2)**
>
> ## Weakness 2
> >The proposed method does not guarantee functionality under arbitrary combinations of preference objectives. In particular, the structure formation phase is fixed to use CLIP as the guiding signal. Therefore, in terms of exploring the trade-offs inherent to true multi-objective preference optimization, the contribution appears somewhat limited.
>
> Thank you for pointing this out. Please allow us to clarify: our method does not have the restriction of "the structure formation stage must use CLIP as the guiding signal." In fact, the phase objectives mentioned in Phase 1 are **all formulated in the form of $F^{str}$**. The core idea of our method is that for each stage, any evaluation metric that can effectively reflect the generation characteristics of that stage can be used, **without the requirement to use any specific metric**. As noted in the appendix on line 806, we explicitly state: *"Note that the framework is extensible to other objectives; see Section 3.4 for ablation studies,"* and experimental validation is provided in the ablation study section (Section 3.4).
>
> During the structural phase, CLIPScore is a widely studied evaluation, representing a broad family of semantic evaluations. In our framework, any metric that reflects structural consistency or semantic layout is acceptable. Therefore, **we employed various alignment evaluation models as alternatives, including (ALIGN, LAION-C, ViT-H)**. As shown in Appendix D.2 of the newly uploaded PDF, we provide new experimental results, which indicate that reasonable changes in the structural objective do not affect the effectiveness of our method. Additionally, to further alleviate your concerns, we have already conducted **replacement experiments for the textural phase and the main objective**, as shown in Fig. 7.
>
> Overall, MultiTune is **not bound to a single CLIPScore**. Instead, it dynamically binds **any** metric that reflects "structural features" in the early generation stages. The purpose of this approach is to align with the natural evolution of diffusion models from global structure to local texture, prioritizing structure-related objectives during the structure phase and then switching to texture-related metrics during the texture phase, achieving a "dynamic optimization" that adapts to the evolving process.
>
> ---
> ## Weakness Response Summary
>
> We greatly appreciate your detailed reading and feedback regarding the two points you raised in the Weakness section. Upon careful review, we realized that, due to the **integration of RL, multi-objective optimization, and diffusion models** in this work, **some relatively basic or trivial technical details may not have been fully elaborated, which led to a slightly different understanding of our original intent than we had anticipated**. However, in essence, our contribution is very much **aligned with what you may have expected**, and we are honored by this. We also apologize for not elaborating on certain foundational concepts, which may have caused some of your concerns. We will further refine the relevant descriptions in the revised version to more clearly convey the core ideas.

---

> ### Author Response · Authors · 2025-11-18
> **Response to Reviewer LokY (Part3)**
>
> ## Question 1-1
> > Could you tell me why CLIP and ES are used solely as feedback signals for exploration rather than being directly optimized?
>
> Thank you very much for your question. We are not quite sure what you mean by "ES," as this seems to be a typo, so we have uniformly understood it as "PS" in the following text. Please feel free to point it out if there is any misunderstanding.
>
> As we clearly stated in the Introduction (lines 43-65) and Related Work (lines 740-762), to avoid the issues arising from mixed optimization of different objectives, each objective in our multi-objective framework is optimized in isolation, that is, we perform separate optimizations for each goal.
>
> Specifically:
> - **RL policy optimization method [1]**: For predefined preference goals (i.e., objectives determined by humans based on the downstream tasks before fine-tuning, such as aesthetics, alignment with human language, etc.), these are the primary optimization targets, and we optimize them using policy optimization techniques.
> - **Exploration-guided optimization**: For phase objectives, to prevent conflict with the policy optimization of the main goal, we adopted an exploration-guided optimization approach (as discussed in the "Mutual Boost" subsection of the main text, lines 216-226). To help you better understand our mechanism, we provide a more detailed explanation here: Specifically, we define the reward for the phase objective, $R^{\text{Phase}}$, as follows:
>
> $R^{\text{Phase}} = \Delta O^{\text{phase}}_t \cdot R^{\text{int}}_t$
>
> In this equation, we calculate the phase objective score difference $\Delta O^{\text{phase}}\_t$  between two adjacent latents $x_{t-1}$ and $x_t$.
>
> If $\Delta O^{\text{phase}}_t$ is positive, then $R^{\text{Phase}}$ is positive (as shown in the equation at line 213,
> $R^{\text{int}}_t$ is always positive, so when $\Delta O^{\text{phase}}_t$ is positive, $R^{\text{Phase}}$ will also be positive).
>
> This indicates that the current step's $x_{t-1}$ has a higher score compared to the previous step $x_t$, suggesting that the current exploratory action is beneficial for improving the phase objective, and thus, the action will be encouraged.
>
> Conversely, if the adjacent latent pair's $\Delta O^{\text{phase}}_t$ is negative, it indicates that the current exploratory action caused a decline in the phase objective, and thus, the action will be suppressed in future iterations. This process demonstrates how phase objectives and exploration actions interact, ensuring that exploration is guided toward improving the phase objective. Therefore, this is an **indirect optimization**.
>
> In summary, the direct optimization of the main objective (predefined preference optimization) through policy optimization is combined with exploration-guided, indirect optimization (of the phase objectives). This approach achieves the decoupling of objectives, preventing the disadvantages of mixed optimization.
>
> **Overall**: MultiTune does not perform policy gradients on phase objectives but instead adopts a differential signal-based shaping approach: when the phase objective increment $R^{\text{Phase}}$ is positive, the current action is considered to push the model in the correct direction for the phase objective and is rewarded; when it is negative, a negative reward is given to prevent deviation. This optimization method **entirely avoids competition between phase objectives and main preference goals** in the same policy gradient channel, thereby **achieving decoupling at the objective level**: the main objective is optimized via PPO, while phase objectives influence exploration behavior through shaping, thus achieving optimization indirectly.
>
> [1] Proximal policy optimization algorithms
>
> ---
>
> ## Question1-2
> > Equation (4) does not appear to represent multi-objective preference learning?
>
> Please refer to the response to Weakness 1

---

> ### Author Response · Authors · 2025-11-18
> **Response to Reviewer LokY (Part 4)**
>
> ## Question 2
> > Why did the authors choose to fix the extrinsic reward to AES instead of dynamically optimizing CLIP or ES depending on the phase change?
>
> Please allow me to clarify this in two parts:
>
> (1)  **Experiments already included in the original manuscript**:
> Please refer to Table 2 in the PDF.
>
> (2)  **Additional experiments specifically added to better address your concerns**:
> Please refer to Sec. 3.4 and Appendix D.3.
>
> As we clarified in our response to the weakness you raised, MultiTune is **not bound to a specific score function**. Instead, it binds any metric that can reflect "structural features" in the early generation stages, and metrics that reflect "texture quality" in the later stages of generation. The goal of this approach is to align with the natural evolution of diffusion models from global structure to local texture, prioritizing the optimization of structure-related objectives during the structure phase, and switching to texture-related metrics during the texture phase. This enables a dynamic optimization process that adapts to the evolving stages.
>
> Therefore, **any scoring model that can reflect structural features and texture quality can be used** within the framework we propose. We have provided experimental results supporting this discussion in the main text. Please refer to the relevant sections for further details (Sec. 3.4 and Appendix D.3).
>
> ---
>
> ## Question 3
> > Have the authors considered alternatives to using CLIP as the guiding signal during the structure formation phase?
>
> Please refer to the response to weakness 2, where we provide results using LAION-C and ALIGN as the alternative for CLIP.

---

> ### Author Response · Authors · 2025-11-19
> **Response to the Reviewer LokY**
>
> Dear Esteemed Reviewer LokY,
>
> We sincerely appreciate your insightful questions and constructive suggestions. We have prepared a detailed and thorough response to all the issues you raised, and we respectfully ask you to review it. Should you have any additional concerns or require further clarification, please do not hesitate to let us know, and we will do our utmost to provide a satisfactory answer.

---

> > ### Author Response · Authors · 2025-11-20
> >
> > Dear Esteemed Reviewer LokY,
> >
> > First, please allow all authors to extend our warm greetings to you. We sincerely appreciate the time and care you have devoted to reviewing our work. In this rebuttal round, we provide detailed responses to your insightful questions regarding our multi-objective formulation and the potential influence of switching evaluation rewards at different stages of training. We have addressed these points thoroughly in the corresponding comments.
> >
> > To offer you a clearer basis for evaluation, we have additionally conducted new experiments specifically tailored to your concerns. **These results—showing the effect of using different evaluation rewards across stages—have been added to the PDF as Fig. 7, and are further discussed in Section 3.4 of the main text and Appendix D.3**.
> >
> > We are deeply grateful for your constructive feedback, which has significantly helped us clarify and strengthen our work.
> >
> > With our highest respect,
> > All coauthor

---

> ### Author Response · Authors · 2025-11-26
>
> Dear Reviewer,
>
> As the discussion deadline is approaching, we sincerely hope that our responses and revisions have helped clarify your concerns. We greatly appreciate the time and effort you have devoted to reviewing our work. If there are any remaining questions, uncertainties, or points you would like to explore further, we would be very glad to continue the discussion and provide more detailed and comprehensive explanations. Please feel free to let us know if there is anything else we can address.

---

### Official Review · Reviewer_eoCG · 2025-11-01

**Soundness:** 1
**Presentation:** 1
**Contribution:** 1
**Rating:** 2
**Confidence:** 3

**Summary:**

The author divides the diffusion process into two objectives dynamically: Phase and Main Objective. The author claimed it reached SOTA performance.

**Strengths:**

## Presentation: ~5th percentile
It can be observed that the grammar and paragraph organisation are nearly flawless; however, anything beyond a single paragraph becomes nonsensical from the perspective of a reader (see below).

## Soundness: 10th~50th percentile
The paper uses experimental results to support its method, but I find it challenging to verify the details (see below).

## Contribution: 5~25th percentile

The divisions of denoising steps may seem arbitrary in theory, but they are reasonable in practice.

## Note:

I hope the AC is aware that the rating is calibrated using estimation of percentiles to reduce evaluation noise effectively.
The rating is simply the mean of the three aspects.

**Weaknesses:**

Based on the writing, this paper should not be accepted.

## Presentation

The writing is almost unintelligible. By the time I finished your methodology:

1. I have a limited understanding of the Phase/Main objective beyond the term you coined, including how it is implemented and the rationale for choosing the key phase transition step $t$ (I assume it’s Eq (3) with 80% confidence, but see the issues below). I was misled by Figure 2 as the caption does not present the details of Figure 2.
2. Is your approach an inference-time scaling method, or a training method with a different objective? I’m about 80% confident it’s the latter, but you shouldn't make me guess.
3. Are you using DDIM/DDPM/score-based SDE to model the diffusion model? I assume it’s DDIM, based on $\hat{x}_0(x_t)$ on Line 173 with 60% confidence. **This is a fundamental detail that should be presented to every reader.**

These details are either missing altogether or scattered across the text rather than presented coherently. As a result, the reading experience is terrible as I have to keep half-formed guesses in mind just to follow along.



### Equation 3

I believe Equation (3) defines the criterion for a phase transition, though it takes more effort to discern this compared to other papers. I still raise some issues.

1. You should group this equation (on page 4) with Figure 1 (on page 2) if Figure 1 is the evidence of this choice of design.
2. Why does the definition on Line 193 call variance (and why does it have no equation labelling?) What does the arrow mean in Equation (3)? What is the indicator for phase transition used for?





## Contribution

### Reproducibility

A lot of important details are oversimplified, especially for those parameter settings. I don’t think a reader can reproduce your result by reading this paper.

### Novelty

The observation that diffusion models denoising in a hierarchical manner is not new, especially since this has already been stated in several theoretical papers like score-based SDE (Yang et. al), EDM (Karras et. al.)

## Soundness

I want to evaluate the soundness based on the implementation details and the methodology. However, my limited understanding of the methodology made it difficult for me to make an accurate judgement, so I can only give my lower bound of my estimation. To be fair, I will lower my confidence to 3. Admittedly, the rather unpleasant experience I had while reading this paper may have influenced my assessment in this respect, although I recognise that I should have evaluated it independently.



The final rating was simply a linear transform of the average of the estimated percentile.

**Questions:**

See Weaknesses.

---

> ### Author Response · Authors · 2025-11-16
> **Our Standpoints toward the Comment from Reviewer eoCG with Evidence (Part1)**
>
> Dear Reviewer eoCG,
>
> First, on behalf of all the authors, we would like to extend our sincere greetings and express our gratitude for the time and effort you dedicated to reviewing our submission. In reading your comments, we sensed a **strong emotional reaction** toward our manuscript, particularly in your statement:
> > “Admittedly, the rather unpleasant experience I had while reading this paper”.
>
> This came as a surprise to us, and we would like to better understand the specific reasons that led to your negative reading experience.
>
> The reason we raise this considerable point is mainly based on the following observations:
> 1. Other reviewers, apart from you, have generally provided more positive assessments for *soundness, presentation, and contribution*.
>
> 2. Your review did not specify **concrete** substantive issues or technical errors, leaving us unable to determine whether the “unpleasant experience” you mentioned arose from the content itself or from your private factors.
>
> As a result of the above observation, we have to further clarify:
> 1. Could it be that certain ideas or viewpoints in this work conflict with interests or perspectives within your research area, potentially influencing your evaluation of the manuscript?
> 2. Or, is it possible that unrelated difficulties or stress you recently encountered might have amplified the negative experience during the reading process？
>
> In any case, we fully respect and understand your feelings, and we apologize for any discomfort this may have caused. At the same time, we sincerely hope to receive specific and actionable suggestions for improvement, so that we can more effectively refine the work. We also wish you all the best and hope that everything is going smoothly on your side.
>
> # Our Position:
> Please allow us to express our viewpoint as follows: We believe that even if there may exist potential conflicts of interest regarding ideas or concepts, or if the reading experience was affected by external circumstances, the evaluation of a scientific work **should still be grounded in its objective contributions rather than any emotional factor**. Maintaining professionalism in commentary is a fundamental premise of academic discourse.
>
> We also noticed that you provided **emotional guidance** to the AC **before the reviewer discussion opened and before seeing the other reviews**; moreover, you explicitly mentioned your “limited understanding of the methodology”, to quote your original words:
> > “However, my limited understanding of the methodology made it difficult for me to make an accurate judgement, so I can only give my lower bound of my estimation.”.
>
> Under such circumstances, it is concerning that you nonetheless attempted to influence the AC’s judgment based on your subjective impressions:
> > “I hope the AC is aware that the rating is calibrated using estimation of percentiles to reduce evaluation noise effectively. The rating is simply the mean of the three aspects.”
>
> This appears **inconsistent** with your simultaneous emphasis that
> > “the rating is simply the mean of the three aspects” and that percentile calibration can effectively reduce noise.
>
> Hence, we have to highlight the issues we observed from your comment, with clear evidence and according explanations as shown in the following part.

---

> ### Author Response · Authors · 2025-11-16
> **Our Standpoints toward the Comment from Reviewer eoCG with Evidence (Part2)**
>
> # Detailed Evidence
>
> We genuinely respect your expertise as a specialist in the AI field, and therefore, we also believe that **emotions should not influence judgment**. However, the clear **logical inconsistencies and contradictions** in your review compel us to question whether some parts of the review may have been generated under time pressure or with AI assistance without careful proofreading. Specifically, we observe the following issues, which we organized into two sections:
>
> ## Section A: Clear contradictions
>
> ### 1. **Inconsistency regarding the evaluation of writing quality**
>
> In the Presentation section, you stated the following to acknowledge the quality of the writing.
> > “the grammar and paragraph organisation are nearly flawless,”
>
>
> However, in the Weakness section, you wrote:
> > “Based on the writing, this paper should not be accepted.”
>
> These two statements directly conflict with each other: within the same review, the writing is described as “nearly flawless,” yet also cited as the primary reason for rejection. **This contradiction makes it difficult for us to understand your actual position**.
>
> ### 2. **Self-contradictory judgment about methodological soundness within a single sentence**
>
> You first describe the method as unreasonable, and then immediately state that it is reasonable, quoting your words:
> > “The divisions of denoising steps may seem arbitrary in theory, but they are reasonable in practice.”
>
> **This abrupt shift does not support any coherent argument** and makes it unclear how you truly assess the method. Considering that similar inconsistencies appear elsewhere in the review, we cannot rule out the possibility that this may be due to AI-assisted drafting without careful verification. As you reminded us, “you shouldn’t let us guess”; likewise, we hope you will not put us in a position where we must guess your actual stance.
>
> ### 3. **Inconsistent confidence in your judgment of this work**
>
> Earlier in your review, you firmly advocated for rejecting the paper and even encouraged the AC to do the same without any specific reason (your original statements include:
> > “Based on the writing, this paper should not be accepted.”;
> > “I hope the AC is aware that the rating is calibrated using estimation of percentiles to reduce evaluation noise effectively. The rating is simply the mean of the three aspects.”
>
> However, in your final summary, you wrote:
> > “However, my limited understanding of the methodology made it difficult for me to make an accurate judgement, so I can only give my lower bound of my estimation.”
>
> Moreover, we do not see any concrete explanation of the “estimation of percentiles,” and the subjective assertions in the earlier parts of your review appear inconsistent with the goal of “reducing evaluation noise.”
>
> **Contradiction**: If you were confident enough earlier to recommend rejection and guide the AC accordingly, why do you later state that you **do not fully understand the methodology**? The earlier statements convey high confidence in rejecting the work, whereas the later admission indicates limited understanding.
>
> ### 4. **Inconsistency regarding fairness in the review**
> You stated that you intend to conduct the evaluation fairly. Yet in the sentence immediately before this, you indicated that because you could not understand the work, you would give a low score:
> > “However, my limited understanding of the methodology made it difficult for me to make an accurate judgement, so I can only give my lower bound of my estimation.”
>
> **Contradiction**: **Claiming a desire for fairness is incompatible with assigning the lowest score precisely because you felt unable to understand the method**. Taking such a definitive negative action under these circumstances appears inconsistent with the principle of fairness you invoked.

---

> ### Author Response · Authors · 2025-11-16
> **Our Standpoints toward the Comment from Reviewer eoCG with Evidence (Part3)**
>
> # Detailed Evidence
> ## Section B: Concerning Issues in the Review
>
> ### 1. **Unreasonable attempt to influence the reviewing process**
> Before the reviewer discussion started and before seeing the assessments of other reviewers, you encouraged the AC to reject the paper. We find this problematic, **especially given your own admission of not understanding the methodology** as :
> > “However, my limited understanding of the methodology made it difficult for me to make an accurate judgement, so I can only give my lower bound of my estimation.”
>
> Under such circumstances, urging the AC toward a negative decision as:
> > “I hope the AC is aware that the rating is calibrated using estimation of percentiles to reduce evaluation noise effectively. The rating is simply the mean of the three aspects.”
>
> appears to be **unfair and may unnecessarily introduce tension among reviewers**.
>
> ### 2. **Inconsistency in following the official review template**
>
> Your review **did not adhere to the structure suggested by the official guidelines**. For instance, the Weakness section was expressed in very general terms, the overall template was incomplete, and you introduced additional, unofficial section titles such as Soundness and Contribution. These extra section headers **appeared multiple times**, yet their content was inconsistent across mentions. This deviation from the template makes it **difficult to clearly interpret your evaluation and your actual intent**.
>
> ### 3. **Unreasonable scoring practice**
> Despite explicitly stating that you did not understand the paper, as your word:
> > “However, my limited understanding of the methodology made it difficult for me to make an accurate judgement, so I can only give my lower bound of my estimation.”
>
> You assigned the **lowest possible score and, without providing meaningful justification**, gave the lowest score for **each sub-aspect** as well. This stands in stark contrast to the assessments of other reviewers who followed the basic norms of peer review.
>
> ### 4. **Logical inconsistencies**
> Your review contains **several instances of internal contradiction**, some occurring even within the same sentence. These issues have already been outlined in Section A.
>
> ### 5. **Issues with language organization**
> There are **frequent self-contradictory, self-answering expressions**, such as:
> > “The divisions of denoising steps may seem arbitrary in theory, but they are reasonable in practice.”
>
> > "Is your approach an inference-time scaling method, or a training method with a different objective? I’m about 80% confident it’s the latter."
>
> This raises concerns about whether **overly simplistic AI assistance may have been used without proper verification**, as we believe that even basic AI tools typically maintain minimal logical coherence. **By the way**, why is your confidence set to a specified **80%**, neither 70% nor 90%? What is your specific method to quantify this value?
>
> # Summary
> Overall, I would like to once again express, on behalf of all the authors, our respect and sincere gratitude for your efforts. **None of the statements above is intended to show any disrespect toward you**; however, we hope you can understand that we do believe your review contains **very serious factors of bias and negative emotion**. We would greatly appreciate it if you could, like the other reviewers, **explicitly list the questions or concerns** you would like us to address, so that we can better resolve your doubts. We trust that you are a responsible scientist, and we look forward to engaging in high-quality academic discussion with you.
>
> ***Please refer to our next message for detailed point-by-point responses to your current questions.***

---

> > ### Comment · Reviewer_eoCG · 2025-11-16
> >
> > As I have already presented in the review:
> >
> > > However, my limited understanding of the methodology made it difficult for me to make an accurate judgement, so I can only give my lower bound of my estimation.
> >
> > Yes, I do not fully understand the methodology due to the presentation, which may be rooted in my lack of expertise, so I only provide the lower bound. I have confidence in rejecting it due to the presentation issue, not because of the potential contribution it could make to the readers. If you fail to deliver your idea, no matter how good it is, it should not be accepted.
> >
> > While I can probably understand this paper using LLM, your average reader may not have the patience to read it unless they use your paper as data instead; I insist on not doing so by reading your paper with my own eyes.
> >
> > The original score will not anchor my final score, so I suggest the author resolve the presentation issue ASAP so that I can reevaluate just in time.

---

> > > ### Author Response · Authors · 2025-11-17
> > > **Point-by-Point Response (Part1)**
> > >
> > > We sincerely appreciate your response. On behalf of all co-authors, we extend our gratitude and highest respect. Once again, we thank you for the time and effort you have devoted to reviewing our manuscript. Our responses to the issues you raised are provided below:
> > > ## Question1
> > > > I have a limited understanding of the Phase/Main objective beyond the term you coined, including how it is implemented and the rationale for choosing the key phase transition step
> > >  (I assume it’s Eq (3) with 80% confidence, but see the issues below). I was misled by Figure 2 as the caption does not present the details of Figure 2.
> > >
> > > The original manuscript **already** addresses this issue:
> > > At the beginning of our method section (Lines 137–140), we explicitly define both the phase objective and the main objective. Furthermore, to elaborate more comprehensively on their respective distinctions and roles, we provide an extended discussion in Appendix C.
> > >
> > > Additional explanation prepared specifically for your inquiry: Given that questions regarding the phase objective and the main objective persist for you, we offer the following further, more detailed clarification:
> > >
> > > ## Q1A-Section A: Technical Explanation
> > >
> > > In reinforcement learning (RL) fine-tuning of diffusion models, researchers typically specify a task objective prior to training—for example, “improving aesthetic quality” or “enhancing text–image consistency.” This objective is implemented through a **reward evaluation model**, which assigns a score after the diffusion process completes and the final image \( x_0 \) is obtained. In our work, the **Main Objective** corresponds to this standard *final reward*, serving as the metric that evaluates how well the generated result aligns with the predefined task.
> > >
> > > However, the denoising trajectory of a diffusion model spans a large number of timesteps, and the Main Objective provides reward only at the last step while all preceding steps receive zero reward. This is a canonical case of **sparse reward**. During the earlier timesteps, the model has no actionable signal indicating whether it is progressing in the correct direction, which leads to:
> > >
> > > - low learning efficiency and slow convergence;
> > > - unstable training dynamics;
> > > - susceptibility to *reward hacking*, in which the model generates anomalous yet high-scoring images solely to maximize the final score.
> > >
> > > To address this central challenge, we introduce **Phase Objectives**, whose functions are:
> > >
> > > 1. to impose distinct intermediate objectives at different stages of the diffusion process, thereby providing an optimizable signal at every timestep;
> > > 2. to match the natural generative progression of diffusion models—transitioning from global structure to textures—by aligning each stage with its most salient characteristics.
> > >
> > > Diffusion generation inherently exhibits a clear phase structure:
> > >
> > > - early stages predominantly determine global structure and spatial layout;
> > > - middle stages progressively synthesize textures and localized detail;
> > > - late stages introduce only minimal refinements.
> > >
> > > Accordingly, we design **Phase Objectives** that transition dynamically across phases:
> > >
> > > - early stages employ rewards emphasizing structure and semantics (e.g., CLIP-based text–image alignment);
> > > - late stages rely on rewards evaluating fine details and aesthetics (e.g., PickScore or ImageReward).
> > >
> > > These Phase Objectives are transformed into **intra-phase reward increments**, thereby supplying a stable and dense optimization signal throughout the trajectory.
> > >
> > > ### Summary
> > > - **Main Objective**: the final task reward defined prior to training; it is applied only to the final image \( x_0 \).
> > > - **Phase Objectives**: stage-specific intermediate objectives designed to mitigate sparse rewards and reward hacking, guiding the diffusion model to make appropriate decisions at every phase.

---

> > > ### Author Response · Authors · 2025-11-17
> > > **Point-by-Point Response (Part2)**
> > >
> > > Besides the technical explanation, we also provide an intuitive explanation of Q1 as follows.
> > > ## Q1A-Section B: Intuitive Explanation
> > >
> > > ### 1. What is the Main Objective?
> > >
> > > When using RL to fine-tune a diffusion model, we typically want the model to “improve” along a predefined direction, for example:
> > >
> > > - making generated images more aligned with human aesthetic preference;
> > > - or making them more consistent with the input text description.
> > >
> > > To achieve this, one usually adopts an evaluation model that, **after** the diffusion model finishes all denoising steps and outputs the final image \( $x_0 $\), assigns a score to this image.
> > >
> > > This score is the **Main Objective**, which indicates:
> > >
> > > > whether the final generated result of the model satisfies the task requirement.
> > >
> > > However, this reward suffers from a critical drawback:
> > >
> > > - it appears **only at the very last step**;
> > > - all preceding steps receive **no reward at all**.
> > >
> > > This is exactly what is called a **sparse reward**. It leads to:
> > >
> > > - the model not knowing what to do during earlier steps;
> > > - great difficulty in attributing the final outcome to specific earlier steps;
> > > - a high risk of **reward hacking**, where the model discovers strange “loopholes” that yield high scores but produce unrealistic or undesirable images.
> > >
> > > ---
> > >
> > > ### 2. Why do we need Phase Objectives?
> > >
> > > Diffusion models generate images **gradually**, step by step.
> > >
> > > Moreover, these steps exhibit **clear phases**:
> > >
> > > - **early phase**: mainly determines the global structure (coarse shapes, layout);
> > > - **middle phase**: introduces textures, colors, and style;
> > > - **late phase**: changes are very small, essentially making final refinements.
> > >
> > > If there is **no reward** for the earlier steps, the model is effectively “working in complete darkness for dozens of steps, only hoping for a sudden piece of feedback at the very end.”
> > >
> > > This naturally causes aimless learning and unstable training.
> > >
> > > ---
> > >
> > > ### 3. What do Phase Objectives do?
> > >
> > > **Phase Objectives** are designed precisely for those intermediate steps where “no one is telling the model what it should be doing.”
> > >
> > > They provide **clear, phase-specific sub-goals** for the model. Furthermore, these sub-goals **change with the denoising phase** of the diffusion model:
> > >
> > > - in the **structure phase** → use CLIP-like metrics to assess whether the global structure and semantics are correct;
> > > - in the **texture phase** → use metrics such as PickScore to assess visual detail and aesthetic quality;
> > > - in the **final refinement phase** → once all scores become stable and no longer improve, the process can be terminated early.
> > >
> > > In this way, each phase has a corresponding optimization direction. The model can thus make steady, purposeful progress instead of wandering without guidance.
> > >
> > > ---
> > >
> > > ### 4. Summary
> > >
> > > - **Main Objective**:
> > >   the final task objective; it only appears at the last step and evaluates the final image.
> > >
> > > - **Phase Objectives**:
> > >   the phase-wise intermediate objectives we design to address sparse rewards and reward hacking, ensuring that the diffusion model has a correct optimization direction at **every** step.
> > >
> > > We hope that this explanation helps clarify the precise meanings and design motivations of our **Main Objective** and **Phase Objectives**. If you still have further questions on this topic, we would be very glad to continue the discussion and provide more detailed explanations.

---

> > > ### Author Response · Authors · 2025-11-18
> > > **Respectful Responses to the Matters You Brought to Our Attention.**
> > >
> > > Dear Reviewer,
> > > Thank you very much for your thoughtful comment and for encouraging us to submit our responses promptly. We have now completed a thorough and careful reply to all points raised in your first-round review, and we respectfully invite you to examine our responses.
> > >
> > > If you have any concerns, questions, or additional suggestions, please let us know—we will spare no effort in addressing them.
> > >
> > > All authors would like to express our deep appreciation for your time, expertise, and commitment to reviewing our work.

---

> > > ### Author Response · Authors · 2025-11-18
> > >
> > > Dear Reviewer,
> > >
> > > We hope this message finds you well. We were very happy by see your recent note:
> > > “The original score will not anchor my final score, so I suggest the author resolve the presentation issue ASAP so that I can reevaluate just in time.”
> > > We fully agree that thorough clarification will help you gain a clearer understanding of our proposed method. In accordance with your request, we have now submitted detailed responses to all the questions and concerns you raised. In addition, we have just uploaded an updated version of the PDF for your review.
> > >
> > > Please feel free to let us know if any further clarification or revision is needed. We truly appreciate your time, patience, and valuable feedback.
> > >
> > > With our deepest respect and gratitude,
> > > The Authors

---

> > > > ### Comment · Reviewer_eoCG · 2025-11-19
> > > > **Reply**
> > > >
> > > > I leave it to you to comment on my tone.
> > > >
> > > > Research suggests that comparison-based methods are less noisy and often less biased [1]. Simply put, a structured formula is more reliable than intuition alone. This method creates a distribution distinct from a standard bell curve. For those falling on the left tail, resentment is naturally triggered, whereas the other tail remains quiet.
> > > >
> > > > While resistance to such mechanical methods is expected, I adopted this approach deliberately to enhance my internal decision-making process. However, given the pushback you reported, I have decided to keep the specific estimations of the three aspects internal. I will continue to use the original metric privately to ensure the consistency of my ratings.
> > > >
> > > > [1] Kahneman, Daniel, Dan Lovallo, and Olivier Sibony. "A structured approach to strategic decisions." MIT Sloan Management Review (2019).
> > > >
> > > >
> > > >
> > > > ## Writing
> > > >
> > > > The order in which you present your materials is crucial for the reader. As a normal human being, I have a limited context size, reading the paper in the order of the flow, maintaining the gist, so that I can retrieve the information whenever needed. An experienced author will compile similar information for readers to access easily. The way you present information, e.g., "how do you divide the timestep followed by DDIM", "what is the actual threshold you use on Equation (3)?" should be properly organised and indexed with a clear hierarchical structure. In my experience, the high-level organisation is more challenging to polish by an LLM than paragraph/sentence-level.
> > > >
> > > > The paper would arguably be more intelligible if the mathematical formulation were omitted. While I choose to evaluate this work by following the principle of charity and ignoring the equations, the mathematical writing remains a concern and could be grounds for rejection.
> > > >
> > > > ICLR demands precise mathematical writing. For readers with a mathematical maturity, the LaTeX math mode is a language for communicating complex ideas with surgical precision. If you choose to present your idea in LaTeX, there are some fundamental common-sense and writing suggestions. I do not guarantee that they are perfect, as the quality depends on the context and the way you write. Some of them can be patched immediately to save this paper and your readers' time and their cognitive load.
> > > >
> > > > Paraphrase the existing work you based on at the beginning of your methods, except for the equation, you should duplicate it as is. This step may seem trivial to the author at first glance, but the information is crucial for every reader, as it establishes a common language between the author and the readers that the subsequent discussion will use. Build your methods based on the symbols in math_commands.tex
> > > >
> > > > Adhere to the notation and terminology presented in the prior works and in the domain; for example, you can consider using [stopping time](https://en.wikipedia.org/wiki/Stopping_time) to indicate the timestep at which you transition from one phase to another. Even for those who do not understand stochastic processes, the term is self-explanatory enough. Ensure your reader understands the symbol you introduce by providing a formal definition.
> > > >
> > > > Provide a clear example for every high-level concept, especially when introducing it for the first time. Your reader needs a concrete idea of how this is implemented. It doesn’t matter if an alternative implementation exists. If so, you can state it later.
> > > >
> > > > Number all the equations. The equation numbers are not only used for referencing by the authors; they are also used by readers to reference equations in discussion, even outside the reviewing process.
> > > >
> > > > ## Soundness
> > > >
> > > > I primarily evaluated the paper on your logical structure when evaluating soundness. The experimental design logically validates the method. But I find it hard to verify if it can be predicted from the theoretical perspective. The choice of DDPO/DPOK as the primary baseline is sufficient for a fair comparison against prior work. The visualisation is rich, so a reader can check the content with ease, had this paper been written with proper text guidance. Unfortunately, the presentation quality buries the reader's ability to appreciate the technical solidity.
> > > >
> > > > 1. In 3.7 Subjective Evaluation, Line 485, which Appendix? I want to check the details of the subjective test. I am mainly curious about the distribution of the samples being evaluated.
> > > > 2. It is hard for a reader to consume Table 2, overall ablation on Line 365, section 3.4 Analysis of MultiTune, because the mathematical symbols do not match the text description.
> > > >
> > > >
> > > >
> > > > ## Contribution
> > > >
> > > > The contribution is moderately good if well-presented. A more continuous exploration of the timestep (rather than an abrupt phase change) could be pursued.

---

> ### Author Response · Authors · 2025-11-17
> **Point-by-point Response (Part3)**
>
> ## Question 2
> > Is your approach an inference-time scaling method, or a training method with a different objective? I’m about 80% confident it’s the latter, but you shouldn't make me guess.
>
> The original manuscript already addresses this issue:
>
> In the abstract, introduction, and method sections, we explicitly describe MultiTune as an “RL-based **fine-tuning** method for diffusion models.” The term **tune** in our method’s name **MultiTune** further makes this intention clear.
>
> Additional explanation prepared specifically for your inquiry:
>
> Thank you for your question. Our approach is **not** a scaling trick applied during inference; rather, it is a complete fine-tuning framework executed during training.
> Specifically, the proposed MultiTune is an RL-based fine-tuning framework for diffusion models. It defines the optimization targets through the introduction of the Main Objective and Phase Objectives, and updates model parameters during training via the corresponding reward signals.
>
> ## Question 3
> > Are you using DDIM/DDPM/score-based SDE to model the diffusion model? I assume it’s DDIM, based on
>  on Line 173 with 60% confidence. This is a fundamental detail that should be presented to every reader.
>
> We appreciate the reviewer’s correction. Fully following prior work such as **DDPO** and **DPOK**, our method primarily adopts **DDIM**. We will make this detail explicit in the final version of the manuscript.
> Specifically, we employ **DDIM with \( \eta = 1 \)**, which is equivalent to **DDPM** and corresponds to the default discrete formulation of the **Score VP-SDE** framework (see **Score SDE** [1], with a detailed discussion provided in Appendix B of the citation).
>
> **[1]** Song, Yang, et al. *Score-based generative modeling through stochastic differential equations.* arXiv preprint arXiv:2011.13456 (2020).
>
> ## Question 4
> > ## Equation 3
> >  I believe Equation (3) defines the criterion for a phase transition, though it takes more effort to discern this compared to other papers. I still raise some issues.
> > 1. You should group this equation (on page 4) with Figure 1 (on page 2) if Figure 1 is the evidence of this choice of design.
> > 2. Why does the definition on Line 193 call variance (and why does it have no equation labelling?) What does the arrow mean in Equation (3)? What is the indicator for phase transition used for?
>
> 1. **Whether Equation 3 serves as the criterion for phase transition:**
>    Yes, your understanding is entirely correct—Equation 3 is indeed the basis for determining phase transitions.
>
> 2. **Regarding the correspondence between Figure 1 and Equation 4:**
>    We sincerely appreciate this constructive suggestion. Your comment is highly valuable. We have added an explicit reference to Figure 1 near Equation 4 in the main text. Please review the updated PDF.
>
> 3. **Regarding the numbering of equations:**
>    In this paper, equations are numbered only when they need to be referenced elsewhere. If an equation is not referenced later, it remains unnumbered. Numbering is added solely for ease of citation.
>
> 4. **Why is the definition in Line 193 referred to as “variance”?**
>    The definition in Line 193,  $\Delta^2 O_t^{\text{phase}}$ represents the **second-order difference** that we use to characterize the magnitude of change in the Phase Objective between adjacent timesteps.
>    To avoid potential misunderstanding, we will revise the term “variance” to the clearer expression **“second-order difference”** in the updated manuscript, and we will formally assign it an equation number as an independent formula.
>
> 5. **Meaning of the arrow notation:**
>    It denotes “tends to” or “converges to.”
>
> 6. **The criterion used to determine phase transitions is Equation 3.**
>    This criterion simultaneously satisfies two conditions:
>    (1) the incremental improvement of the phase objective approaches zero;
>    (2) the slope of its change between adjacent timesteps likewise approaches zero.
>
>    This indicates that the optimization benefit of the current phase has been essentially exhausted—not only has the improvement of the objective nearly vanished, but the trend of improvement has also disappeared. In other words, continuing to optimize under the current phase objective would no longer yield meaningful gains, and thus the system should adaptively transition to the next phase.
>
> *We sincerely hope that these responses address your concerns. We also welcome any further discussion at any time, and we will provide additional clarification as promptly as possible.*

---

> ### Author Response · Authors · 2025-11-17
> **Point-by-point Response (Part 4)**
>
> ## Question5
> > A lot of important details are oversimplified, especially for those parameter settings. I don’t think a reader can reproduce your result by reading this paper.
>
> Thanks for your comment. Due to space limitations, we provided a brief overview of the experimental setup in **Section 3.1** of the main text. Then, in **Appendix D.1 and Table 6**, we have provided a comprehensive list of all the parameters. Additionally, we have included the **complete main process code** in the supplementary materials. We believe that the information provided is sufficient to assist readers in reproducing the work.
>
> ***If you have any further specific questions regarding reproduction, we are more than happy to assist and cooperate promptly.***
>
> ## Question 6
> > The observation that diffusion models denoising in a hierarchical manner is not new, especially since this has already been stated in several theoretical papers like score-based SDE (Yang et. al), EDM (Karras et. al.)
>
> Thank you very much for your insightful comments. We completely agree that the hierarchical evolution from structure to detail during the generation process of diffusion models has been well-discussed in many previous theoretical works. Therefore, we did not consider the “phased nature of diffusion models” as an innovation in this paper.
>
> Our core contribution lies in
> 1. We propose a mechanism that can adaptively perceive this evolving generative process, which embeds optimization objectives that match the characteristics of each phase.
> 2. We perform multi-objective RL optimization in a *phase-decoupled* manner.
>
> To the best of our knowledge, this RL method, which integrates and decouples multiple preference objectives with the natural evolutionary process of diffusion models, has **not** been explored in prior works.

---

> ### Comment · Reviewer_eoCG · 2025-11-19
> **Miscellaneous**
>
> 1. By Equation Line 177 (Please add an equation labeling) and Equation 2, I can deduce that
> $\Delta^2 O^{\mathrm{phase}}_t = F(\hat{x}_0(x_t))  - F(\hat{x}_0(x_{t + 2}))$. Is my deduction correct? If so, as long as the number of timesteps grows, one of the \rightarrow seems redundant.
> 2. In this paper, x is typically a vector; use [bold font](https://arxiv.org/html/2410.12672v1).
> 3. Equation 3 should be replaced by some alternative presentation. If you still decide to rewrite based on the current formulation, at least you should not use rightarrow (as it is very similar to $\to$) even if it means “tend to”. My time was wasted because the misleading symbol has another rigorous mathematical meaning. $\epsilon_{\mathrm{zero}}$ should be inside equation 3, explictily or implicitly by expanding the symbols. Use [the wedge family](https://tex.stackexchange.com/questions/23792/logical-and-character-in-tex-%E2%8B%80) to indicate logical and, or use set intersection if you formulate in probability space.

---

> > ### Author Response · Authors · 2025-11-19
> > **Response to Esteemed Reviewer eoCG**
> >
> > Dear Esteemed Reviewer eoCG,
> >
> > Thank you very much for your response. From your comments, we can clearly see that you are a highly rigorous AI scientist. Indeed, there is currently no unified or authoritative standard for mathematical notation in our field, and many classic papers do not strictly distinguish between vectors and scalars, largely to maintain accessibility for readers without a strong mathematical background.
> >
> > We fully understand your concerns regarding notation. We also have a solid foundation in notation conventions, such as using boldface for vectors and regular font for scalars. We truly appreciate your attention to these details and fully respect your rigorous standards.
> >
> > We will revise and unify the notation throughout the paper based on your suggestions to ensure it aligns with your expectations. In addition, if any previous misunderstanding caused you discomfort, please accept our sincere apologies. We will share the updated PDF with you as soon as it is ready.

---

> > ### Author Response · Authors · 2025-11-22
> > **Summary of the responses to the followup issues raised by the esteemed reviewer eoCG.**
> >
> > ## **Overall:**
> >
> > We hope that, based on the detailed explanations of our innovations, experimental validation, and the updates made to the PDF, you can reconsider the contribution score. If you are satisfied with the revisions, we kindly ask if you could provide a higher evaluation. If there are any further questions or concerns, we would be very happy to address them and engage in further discussion.
> >
> > Thank you once again for your time and thoughtful review.

---

> > ### Author Response · Authors · 2025-11-24
> > **The response list  in the reviewer’s Miscellaneous section.**
> >
> > For your convenience in locating the questions you raised in the Miscellaneous section, we have prepared a response list to help you easily find the corresponding clarifications:
> >
> > 1.Regarding the formula label issues
> >
> > please refer to ***Items 1 and 2 in Further Clarification (Part 1) as well as the updated full PDF***.
> >
> > 2. Regarding your comment:
> > “By Equation Line 177 and Equation 2, I can deduce that $\Delta^2 O^{\mathrm{phase}}_t = F(\hat{x}_0(x_t)) - F(\hat{x}0(x{t+2}))$. …”
> >
> > We have addressed this ***in Item 3 of Further Clarification (Part 1)***.
> >
> > 3.Regarding your comment:
> > “In this paper, x is typically a vector. …”
> >
> > Our response can be found in ***Item 1 of Further Clarification (Part 1)***.
> >
> > 4.Regarding your comment:
> > “Equation 3 should be replaced by some alternative presentation…”
> >
> > Our response is provided ***in Item 5 of Further Clarification (Part 1)***.
> >
> > 5. For detailed explanations regarding the **contributions and presentation**
> >
> >  please refer to Further ***Clarification (Part 2)***.

---

> ### Author Response · Authors · 2025-11-22
> **Further Clarification for the Followup Issues Raised by the Esteemed Reviewer eoCG (Part 1)**
>
> Dear Reviewer eoCG,
>
> We are truly grateful for the opportunity to clarify the misunderstanding, and we sincerely appreciate your patience. Some of the issues, such as the inconsistency in labeling equations, were due to a lack of a strict industry standard, which led to some confusion. For example, the reason some formulas had labels and others didn’t was that we added equation labeling only to those that needed to be referenced or emphasized within the text. We apologize for any misunderstanding this caused, and we assure you that it was never due to a lack of care or rigor on our part. Your feedback has made us even more appreciative of the thoroughness you bring to your review process.
>
> Now, please allow me to address your new questions and provide the corresponding updates to the PDF, as per your request. Once again, we would like to thank you for the time you have dedicated to reviewing our work.
>
> ---
>
> ### 1. **Regarding your suggestion to distinguish between vectors and scalars:**
>
> We have updated the entire document accordingly. Please review the changes in the PDF.
>
> ---
>
> ### 2. **Regarding your request to add equation labeling:**
>
> We have made the necessary modifications in the PDF by labeling the relevant formulas. Additionally, to make the review process more convenient for you, we have changed our approach and now label all formulas that need to be referenced in the text. Please review the updated document.
>
> ---
>
> ### 3. **Regarding your deduction:**
> > "By Equation Line 177 (Please add an equation labeling) and Equation 2, I can deduce that $\Delta^2 O^{\mathrm{phase}}_t = F(\hat{x}_0(x_t)) - F(\hat{x}_0(x_{t+2}))$. Is my deduction correct? If so, as long as the number of timesteps grows, one of the \(\rightarrow\) seems redundant.”
>
> We kindly ask for your permission to revisit the derivation process here (we mean no disrespect at all). Upon further inspection, we found an algebraic mistake in your deduction that led to a misunderstanding. To ensure accuracy, we discussed this with several other researchers in the field, and they confirmed that the intermediate term $-2F(\hat{x}_0(x_{t+1}))$ should not be eliminated.
>
> Allow us to clarify the correct algebraic process:
>
> - Based on our definitions (Eq. (2), (3), (4), (5)):
>
>   - **Phase 1 / 2**:
>
>    $\Delta F_t = F(\hat{x}_0(x_t)) - F(\hat{x}_0(x_{t+1}))$
>
>   - **Phase Objective**:
>
>   $\Delta O^{\mathrm{phase}}_t = \Delta F_t$
>
>   - **Second-Order Change Definition**:
>
>   $\Delta^2 O^{\mathrm{phase}}_t = \Delta O^{\mathrm{phase}}_t - \Delta O^{\mathrm{phase}}\_{t+1}$
>
>   When we expand the above expression, it becomes:
>
>
>
>   $\Delta^2 O^{\mathrm{phase}}_t$
>  = $\[F(\hat{x}\_0(x_t)) - F(\hat{x}\_0(x\_{t+1}\)\)\]$ \- $[F(\hat{x}\_0(x_{t+1})) -F(\hat{x}\_0(x\_{t+2}))]$
>
>
>  = $F(\hat{x}\_0 (x_t))$ - 2 $F(\hat{x}\_0(x\_{t+1}))$ + $F(\hat{x}\_0(x\_{t+2}))$
>
>
>   However, your deduction was:
>
> $\Delta^2 O^{\mathrm{phase}}_t = F(\hat{x}_0(x_t)) - F(\hat{x}_0(x\_{t+2})),$
>
>   Here, the intermediate term **$\(-2F(\hat{x}_0(x\_{t+1}))\)$** was missed.
>
> May I respectfully ask if you could kindly double-check the algebraic derivation once again? We would like to emphasize that we mean no disrespect whatsoever. We hold your expertise in high regard, and our intention is only to clarify the issue. We truly hope you understand and would appreciate your review of this point.
>
> ---
>
> ### **4.  The symbol issue**
>
> > In this paper, $x$ is typically a vector; use bold font.
>
> We have revised the entire document to reflect the use of bold font for vectors, as per your request. Please review the changes in the updated PDF.
>
> ### **5. Formulation**
>
> Actually, we directly use the `numpy.isclose()` function in Python, which is equivalent to "tend to." Specifically, its default value for $\epsilon_{\text{zero}} = $1e-5. In the revised PDF, we have included a link to the appendix next to the relevant formula, where we provide the **detailed implementation of this issue**.
>
> (*To be continued in the followup part*)

---

> ### Author Response · Authors · 2025-11-22
> **Further Clarification for the Followup Issues Raised by the Esteemed Reviewer eoCG (Part 2)**
>
> ### **6. Regarding the Contribution Scoring:**
>
> We respectfully request that you reconsider the contribution score, given the three major innovations and extensive experimental validation in our work:
>
> #### (1) **Multi-Objective Switching Strategy:**
> We propose a multi-objective switching strategy that aligns with the structure and details of the diffusion process, dynamically focusing on decoupling and optimizing specific objectives at different stages. This approach avoids multi-objective optimization conflicts and allows independent optimization at each stage.
> This is a novel contribution, especially in the context of diffusion models, where such task switching has not been extensively explored before.
> **Evidence**: Fig. 1 (visualization experiment), Fig. 3 and Fig. 7 (multi-objective optimization experiments).
>
> #### (2) **Adaptive Mechanism for Exploration and Utilization:**
> We introduce an adaptive mechanism based on image quality changes that dynamically balances stage and main objectives, enabling real-time focus shifts between exploration and exploitation.
> This addresses a long-standing challenge in RL—balancing exploration and exploitation, which is particularly aggravated when RL is applied to diffusion models due to sparse rewards and long-sequence decision-making.
> We believe this is a significant advancement.
> **Evidence**: Fig. 5 and Fig. 19 (exploration–exploitation balance validation).
>
> #### (3) **Lightweight Multi-Objective Training with Early Stopping:**
> By discarding invalid samples and stopping redundant denoising steps early, we achieve lightweight multi-objective training. Our early stopping mechanism is not a simple heuristic but a two-pronged evaluation based on denoising completion and reward gain. This reduces computational costs while maintaining high-quality results.
> **Evidence**: Tab. 2 (computation overhead comparison), Fig. 6 (time overhead comparison), and comparison with SOTA methods showing that our method requires less computational power for the same score.
>
> ---
>
> ### **7. Extensive Experimentation:**
>
> We believe the experimental validation in our paper is comprehensive and exceeds that of many existing works. We have conducted a total of **20 different types of experiments**, which include:
>
> 1. **Motivation Justification**: Fig. 1 (visual experiments).
> 2. **3D Pareto Representation**: Fig. 3 (comparison with multiple multi-objective optimization methods and RL fine-tuned diffusion models).
> 3. **Backbone Replacement Experiment**: Tab. 1 (shows backbone replacement does not affect method superiority).
> 4. **Cross-Reward Generalization Experiment**: Fig. 4 (proves the generalization capability across different reward metrics).
> 5. **Comparison with Single-Objective SOTA**: Fig. 5 (direct comparison with SOTA single-objective methods).
> 6. **Computational Power Experiment**: Fig. 6 (time overhead for achieving the same score).
> 7. **Stage Goal Swap Experiment**: Fig. 7 (verifying the flexibility of swapping stage objectives).
> 8. **Ablation Study**: Tab. 2 (ablation study results).
> 9. **Visualization of Diversity in Generated Trajectories**: Fig. 8 (exploration encouraged by internal reward).
> 10. **Generalization on Complex Prompts**: Fig. 9 (performance on complex prompts).
> 11. **Alignment Validation**: Fig. 11 (alignment for specific requirements such as color and count).
> 12. **Diversity in Generation**: Fig. 11 (visualizing the diversity of generated results).
> 13. **Human Evaluation Experiment**: Fig. 10 (human evaluation results).
> 14. **Generalization on Complex Datasets (Training)**: Tab. 4 (performance on more complex datasets).
> 15. **Inference on Complex Datasets**: Tab. 3 (inference results on complex datasets).
> 16. **Advanced Backbone Training**: Tab. 5 (results on larger backbones like SDXL).
> 17. **2D Pareto Experiment**: Fig. 12 (2D Pareto front).
> 18. **Structural Metric Change Experiment**: Verification of the adaptability of structural indicators.
> 19. **GenEval and Pick-a-Pic Datasets**: Fig. 14 (performance on additional datasets).
> 20. **Fine-tuning on SD3 Model**: Performance on the latest SD3 model.
>
> We firmly believe these extensive experiments validate the robustness and generality of our method.
>
> ---
>
> ### **7. Regarding Presentation Score:**
>
> We have addressed all formatting and labeling concerns, including bolding vector symbols and adding equation labels, as per your request. Please review whether these updates meet your expectations.

---

> ### Author Response · Authors · 2025-11-24
> **Overall Response to the Reviewer's Miscellaneous**
>
> Dear Reviewer,
>
> We respectfully believe that all the issues raised in the **Miscellaneous** section have been fully addressed. Should you have any further questions, we would be more than happy to provide additional clarification. If you feel that your concerns have been resolved, we would sincerely appreciate it if you could kindly consider updating your score.

---

> ### Author Response · Authors · 2025-11-26
> **Further Discussion on the Validity of the First Point in the Miscellaneous**
>
> Dear Esteemed Reviewer eoCG,
>
> Regarding your comment: "By Equation Line 177 (Please add an equation labeling) and Equation 2, I can deduce that $\Delta^2 O^{\mathrm{phase}}_t = F(\hat{x}_0(x_t)) - F(\hat{x}_0(x\_{t+2})),$
>
> We have provided a detailed response in our rebuttal. **We would like to confirm whether this was a computational slip on your side, or if we might have misunderstood your intended meaning**?
>
> If you have time to reply, we would greatly appreciate your clarification.
>
> Thank you sincerely for your attention.

---

> > ### Comment · Reviewer_eoCG · 2025-11-27
> >
> > I have re-read the paper and found that most of my concerns regarding the presentation have been resolved, although there is still room for improvement. Unfortunately, I can no longer evaluate this work as an uninformed reader.
> >
> > The paper is sound per se, supported by robust experiments that demonstrate its significance. However, the novelty is somewhat dry; it represents an incremental modification of the time parameter rather than a paradigm shift.
> >
> > Equation 6, still, flies in the face of formal notation and triggers an immediate negative reaction in my System 1; consequently, I am assigning a low confidence score to this rating.

---

> ### Author Response · Authors · 2025-11-27
>
> We are truly grateful for the support（Raise our score to 6） and commitment you have shown to our work. Your profound understanding of the mathematical expressions reflects your rigorous attitude toward scientific research. We will continue to improve the PDF based on all your earlier and most recent suggestions.
>
> Lastly, on behalf of all authors, please allow me to express our heartfelt thanks for your time and contributions.

---

### Author Response · Authors · 2025-11-30
**Overall Comment (3): Point-by-Point Response List for AC**

Dear AC,

To help you quickly verify that we have addressed all reviewers’ concerns, we provide below a structured summary of our responses, organized into three parts: **comment replies, PDF revisions, and experiments**.

---

## Reviewer: eoCG ✅
All **9 weaknesses** have been fully addressed. The reviewer increased the score **twice** before the rollback, as visible in their replies.

- On **Nov. 19**, we resolved the initial concerns and pointed out the reviewer’s technical error, resulting in the **first score increase**.
- Please note that in our initial response, all references to “Part” correspond to sections in the **Point-by-Point Response**.

### Main Weaknesses (for the first score raise)

- **W1: Main objective vs. stage objectives**
  *Comment:* Part 1, Part 2

- **W2: Inference-time scaling vs. training**
  *Comment:* Part3_Q2

- **W3: DDIM or DDPM**
  *Comment:* Part3_Q3

- **W4: Understanding Equation (3)**
  *Comment:* Part3_Q4

- **W5: Hyperparameter explanation**
  *Comment:* Part3_Q5
  *PDF:* Sec. 3.1, Appendix D.1, Tab. 6
  *Code:* See attachment

- **W6: “Diffusion in a hierarchical manner is not new”**
  *Comment:* Part3_Question6

### Additional Issues (led to the second score increase)

- **Miscellaneous1 & Miscellaneous2: Vector vs. scalar, formula labeling**
  *Comment:* Part1 (1, 2, 4)
  *PDF:* Updated accordingly throughout the paper

- **Miscellaneous3: Redundant arrows / clarification of “tend to”**
  *Comment:* Part1 (3, 5)

⚠️ **Note:** The reviewer made an **algebraic mistake** that led to misunderstanding (further discussed in Part1_3).
  We corrected it, and the reviewer raised no objection in the following reply.

---

## Reviewer: LokY ✅
All weaknesses and questions have been fully addressed. ***All reviewer LokY’s requested new experiments have been added***. No response was received despite multiple follow-ups.

### Weaknesses


- **W1: “Equation (4) for multi-objective task?”**
  *Comment:* Part1
  ⚠️ **Note:** The misunderstanding stemmed from lack of familiarity with RL.

  ⚠️ **Note:** Due to the strong request from Reviewer eoCG, we have now numbered all equations; as a result, the previous Equation (4) has been renumbered as Equation (8).

- **W2: “The method is not guaranteed under arbitrary preference combinations.”**
  *Comment:* Part2
  *PDF:* Appendix line 806, Sec. 3.4 (ablation study), Appendix D.2
  *Experiments:* Fig. 7 and Tab. 2 (original), plus new Fig. 13
 ⚠️  **Note:** The reviewer appears to have overlooked explanations and experiments already present.

### Questions

- **Q1: Direct optimization and relation between Eq. (4) [Eq. (8)] and multi-objective learning**

  *Comment:* Part3

 - **Q2 & Q3: Equivalent to W2**
  *Comment:* Part4
  *Experiments:* Fig. 7, Tab. 2 (original), and new Fig. 13

---

## Reviewer: d9rJ ✅
All weaknesses and questions have been fully addressed. ***All reviewer d9rJ’s requested new experiments have been added***. No response was received after multiple reminders.

### Weaknesses

- **W1: Need for complex prompt sets (in particular, GenEval)**
  *Comment:* Part1_W1
  *Already in paper:* HPSv2 and ImageNet-A (Tab. 3, Tab. 4)
  *Added for clarity:* New results on **GenEval** and an additional experiment on **Pick-a-Pic** (Fig. 14).

  ⚠️ **Note:** The reviewer did not notice the existing results.

- **W2: More details on computational cost**
  *Comment:* Part1_W2
  *Experiments:* Tab. 2 and Fig. 6

### Questions

- **Q1: How are the scores in Fig. 1 computed? Which models are used?**
  *Comment:* Part1_Q1

- **Q2: How is the stage-switching timestep defined?**
  *Comment:* Part2_Q2

- **Q3: Section X.X.**
  Typo acknowledged and fixed in the PDF.

---

## Reviewer: zFFf   ✅
***All reviewer zFFf’s requested new experiments have been added***. All weaknesses and questions have been fully addressed, and we received strong positive feedback from this reviewer.

### Weaknesses

- **W1: Whether the method applies to DiT-based models**
  *PDF:* Added discussion in Appendix D.6
  *Experiments:* New results in Fig. 15 and Tab. 7

- **W2: Additional discussion for Tab. 5**
  *PDF:* Added detailed analysis highlighted in blue in Sec. 3.6

### Questions

- **Q1: What is “stage reward”?**
  *Comment:* Q1
  *PDF:* Appendix D.2
  *Experiments:* Fig. 7, Tab. 2, and newly added Fig. 13

- **Q2: Section X.X.**
  Typo fixed in the updated PDF.

- **Q3: Broken content in the PDF**
  Display issues across different browsers have been fixed.

We sincerely appreciate the reviewer’s continued engagement. After we completed the additional SD3 (DiT-based) experiments following the reviewer’s guidance, the reviewer further raised the score and explicitly stated being *“convinced of the soundness and impact of this work.”*

---

### Author Response · Authors · 2025-11-30
**Overall Comment (2): Experiment List for AC**

⚠️  Dear AC, Since we revised the paper multiple times in response to reviewer comments and discussions, the numbering of figures and tables has changed compared to the original comments and rebuttal messages. We kindly ask you to refer to **this list** as the authoritative mapping between experiments (Fig. Sec. Tab.) and reviewer queries. Thank you for your understanding.

---

# Experiment List

Here we provide a detailed list of experiments in the paper and how they address the reviewers’ comments:

1. **Motivation Justification**
   Fig. 1 — Visual experiments supporting the design motivation.

2. **3D Pareto Representation**
   Fig. 3 — Comparison with multiple multi-objective optimization methods and RL-fine-tuned diffusion models.

3. **Backbone Replacement Experiment**
   Tab. 1 — Demonstrates that changing the backbone does not affect the superiority of the proposed method.

4. **Cross-Reward Generalization Experiment**
   Fig. 4 — Verifies generalization across different reward metrics.

5. **Comparison with Single-Objective SOTA**
   Fig. 5 — Direct comparison with state-of-the-art single-objective methods.

6. **Computational Cost Comparisons**
   Fig. 6 — Runtime comparison under the same target score.
   ✅ *Addresses: Reviewer d9rJ – Weakness 2*

7. **Computational Overhead per Component**
   Tab. 2 — Time and memory consumption for each proposed component.
   ✅ *Addresses: Reviewer d9rJ – Weakness 2*

8. **Stage Goal Swap Experiment**
   Fig. 7 (Phase 2 & Main) and Fig. 13 (Phase 1) — Validates the flexibility of swapping stage objectives.
   ✅ *Addresses: LokY – W2, Q1, Q3; d9rJ – Q1; zFFf – Q1*

9. **Ablation Study**
   Tab. 2 — Ablation results of different modules.

10. **Visualization of Trajectory Diversity**
    Fig. 8 — Demonstrates that intrinsic rewards encourage exploration.

11. **Generalization on Complex Prompts**
    Fig. 9 — Performance on complex linguistic prompts.

12. **Alignment Validation**
    Fig. 11 — Verification on alignment requirements such as color and object count.

13. **Diversity in Generated Results**
    Fig. 11 — Visualization of generation diversity.

14. **Human Evaluation**
    Fig. 10 — Human preference study.

15. **Training on Complex Datasets**
    Tab. 4 — Performance on more complex datasets during training.
    ✅ *Addresses: Reviewer d9rJ – W1*

16. **Inference on Complex Datasets**
    Tab. 3 — Results of inference-time evaluation on complex datasets.

17. **Advanced Backbone Evaluation**
    Tab. 5 — Results on larger backbones (e.g., SDXL).

18. **2D Pareto Front Experiment**
    Fig. 12 — Two-dimensional Pareto analysis.

19. **Additional Datasets: GenEval & Pick-a-Pic**
    Fig. 14 — Evaluation on additional benchmarks.
    ✅ *Addresses: Reviewer d9rJ – W1*

20. **Fine-tuning on SD3 (DiT-based Model)**
    Fig. 15 Tab. 7 — Performance on the latest SD3 architecture.
    ✅ *Addresses: Reviewer zFFf – W1*

21. **Results on the Base Backbone**
    Tab.5 — Performance without any design.
    ✅ *Addresses: Reviewer zFFf – Q2*

---

This experiment list is also cross-referenced in the **Experiment List** in the overall comment to facilitate quick lookup.

---

### Author Response · Authors · 2025-11-30
**Overall Comment (1): Rebuttal Summary for AC**

Dear AC,

Please allow us to summarize our rebuttal efforts to help save your time. Briefly speaking, we are confident that **we have addressed all reviewers’ questions without any omission** (detailed evidence can be found in the dedicated _issue–solution index_ and the exact location of each response provided for you at **overall comment (3)**), the detailed experimental list at **overall comment (2)**:

---

## 1) Experimental Results: Clarifications and Extensions

Based on the **16 experiments** already included in the original paper, we provided further clarifications and additionally conducted **5 new independent experiments** (bringing the total to **21 experiments** in the revised version). The key additions include:

a. **Phase-1 objective replacement experiments**
   Comparing **CLIP** with **ALIGN** and **LAION-C** – see **Fig. 13**.

b. **More detailed computational cost analysis**
   Including time and memory usage for each proposed component – see **Tab. 2**.

c. **Evaluation on broader datasets**
   Including **GenEval** and **Pick-a-Pic** – see **Fig. 14**.

d. **Additional experiments on a DiT-based model (SD3)**
   Results are shown in **Fig. 15** and **Tab. 7**.

For the complete list of experiments and the corresponding reviewer questions they address, please refer to **overall comment (2): Experiment List**. Through these additional clarifications and experiments, we have addressed **all reviewers’ concerns from the empirical perspective**.

---

## 2) Clarifications on Innovation, Theory, and Design

Building on the existing discussion of multi-objective optimization and core contributions in the paper, we further expanded on the following aspects:

a. **The novelty of decoupled multi-objective learning**
   We propose the **first** method that leverages the multi-phase nature of diffusion models to **decouple** multi-objective optimization.
   Please see: _Point-by-point response to eoCG Part 1 and Part 2_, and _LokY Part 2_.

b. **Theoretical rigor and methodological background**
   Including the theoretical foundation of the scheduler used in our work.
   Please see: _eoCG Part 1_.

c. **Clarification of objective switching criteria and optimization targets**
   Please see: _d9rJ Part 1_ and _LokY Part 1_.

For a complete directory of all responses and the corresponding reviewer issues, we provide a full index in **overall comment (3): Point-by-Point Response List**, which allows you to precisely locate each reviewer’s questions and our detailed answers. Through these discussions, we have addressed **all theoretical concerns** raised by the reviewers.

---

## 3) Reviewers’ Attitudes and Outcomes

Here we summarize each reviewer’s evaluation and interaction with us:

-  ✅ **eoCG** (**Support Accept after Discussion**):
  In the initial review, this reviewer had **certain misunderstandings** and concerns about our work. After our detailed clarifications on the method design and careful verification of each raised issue, the reviewer ***increased the score twice*** (please refer to the **Official Comment by Reviewer eoCG**)  during the discussion phase and eventually **acknowledged our work**.

- **LokY** (**No Response After Multiple Updates**):
  The reviewer initially gave a score of **332**. After we fully addressed all concerns (please see the list in overall comment (3)), *we have **not** received any further response despite **multiple** updates*.

-  **d9rJ** (**No Response After Multiple Updates**):
  The reviewer initially gave a very strong score of **333**. After we resolved all the issues during the rebuttal phase (please see the list in overall comment (3)), *no further response has been received after **multiple updates***.

-   ✅ **zFFf** (**Strongly Support Accept After Further Acknowledgemnet**):
  This reviewer showed a positive attitude toward our work from the beginning of the discussion. During the rebuttal, with guidance from the reviewer, we made a substantial effort to reproduce our method on one of the most advanced models, **SD3 (DiT-based)**. Subsequently, the reviewer further **increased the score and highly endorsed our work**  (please refer to the **Official Comment by Reviewer zFFf**) , stating that they were  ***"convinced of the soundness and impact of this work"*.**  This strong recognition has been a great encouragement to us.

---

## Conclusion

In summary, we believe that **all concerns have been thoroughly resolved**. In particular, the multiple score increases from **eoCG** and **zFFf** after in-depth discussions further reinforce our confidence that our responses have effectively addressed the reviewers’ doubts. If you have any remaining questions, we would be more than happy to discuss further.

We will next provide the **experiment list** and the **point-by-point response list** for your reference.

Thank you very much for your time and consideration.

---

### Meta-Review · Area_Chair_rBf8 · 2026-01-07

**Summary:**

Some of the reviewers cited presentation issues and/or unclear methodological/implementation details. Although some of these were addressed during the discussion phase, from my read of the paper, I think it's difficult to tell whether the empirical performance differences between the proposed method and the baselines (DDPO, MORL, DAS, etc) are due to applying different objectives at different denoising phases, or due to the implementation of RL algorithm or other implementation differences. The only place that mentioned how optimization is done is Algorithm 1 Line 26 "optimizing final reward via PPO" in Appendix C. There is no mentioning of the RL algorithm implementation is aligned with any of the baselines.

Now, the question is -- Are there any ablation studies done to study this in a fair comparison setting? Based on the writing and discussion, I guess Table 2 might be what I'm looking for, but there is no explanation/description for how to read Table 2 in the caption nor in main text. It seems like Table 2 row 1 means that optimizing against intrinsic reward alone can achieve better AES score and comparable CLIP scores compared to baselines like DDPO or MORL shown in Table 1 SDv15 columns (Comparing the "Ours" numbers, I can see that Table 2 is probably based on SDv15 though it's not explained). If this is true, the differences shown in Table 1 is more likely to come from implementation differences, not the proposed reward structure.

At this point, I'm certain that the paper doesn't provide enough clarity for me to access the empirical results. Although the core idea of the paper makes sense to me, the execution and presentation quality does not seem sufficient to be accepted at this moment.

**Reviewer Concerns:**

Addressed:
* Empirical Scalability: In response to reviewer zFFf, the authors successfully extended their method to the industrial SOTA SD3 (DiT-based) model, which convinced the reviewer of the work's soundness and impact.
* Dataset Diversity: To satisfy reviewer d9rJ, new experiments were conducted on complex datasets, including GenEval and Pick-a-Pic.
* The authors provided a detailed breakdown of time and memory usage for each component, addressing concerns about the overhead of adaptive mechanisms.
* Objective Flexibility: Authors demonstrated that the framework is not fixed to specific rewards like CLIP, providing experiments with alternatives like ALIGN and LAION-C to show generalizability.

Outstanding:
* Mathematical Formalism: Reviewer eoCG maintained that the mathematical writing remained a concern about Equation 6.
* The presentation was still described as difficult to follow.

**Reviewer Scores:**

Based on the authors' claim, the scores pre-rollback were 8, 6, 4, 4. It was quite diverged.

---

### Decision · Program_Chairs · 2026-01-26

Reject